# The Functioning of Erosion-channel Systems of the River Basins of the South of Eastern Siberia

**Olga I. Bazhenova [1],\*, Aleksandr V. Bardash [1], Stanislav A. Makarov [1], Marina Yu. Opekunova [1], Sergei A. Tukhta [1] and Elizaveta M. Tyumentseva [2]**

[1] V.B. Sochava Institute of Geography SB RAS, Irkutsk 664033, Russia; bazhenova_o49@mail.ru (A.V.B.); makarov@irigs.irk.ru (S.A.M.); opekunova@irigs.irk.ru (M.Y.O.); varitan@yandex.ru (S.A.T.)

[2] Geographical Department, Pedagogical Institute of Irkutsk State University, Irkutsk 664033, Russia; tumencev@irk.ru

\* Correspondence: bazhenova@irigs.irk.ru; Tel.: +7-3952426920

**Abstract:** We revealed the regional features of the functioning of the erosion-channel systems of the Angara, Upper Lena, Selenga, and Upper Amur basins in the south of Eastern Siberia and examined the action of sloping non-channel, temporary, and permanent channel water flows, and presented the patterns of the spatial distribution of soil and gully erosion belts. The development conditions and factors of fluvial processes are considered and the role of cryogenic processes in the increasing activity of water flows is emphasized. The interdecadal dynamic cycles of the erosion-accumulative processes are revealed. A quantitative assessment of soil loss from erosion on agricultural land in the forest-steppe basins was carried out. We made an assessment of the plane deformation of the upper course of the Lena river (Siberian platform) and Irkut (Baikal rift zone and the Irkutsk-Cheremkhovo plain) using cartographic sources of different times, aerial photographs, and satellite imagery. The contribution of extreme fluvial events to sediment redistribution in river basins is shown. Particular attention is paid to the mudflow impact, floods, and channel deformations on the ecological state of the basin systems.

**Keywords:** river basins; soil erosion; gullies; channel processes; mudflows; Eastern Siberia

## 1. Introduction

River basins are the main elements of spatial organization of the landscape geosphere [1]. In this context, the basin approach has become central in dealing with many fundamental and applied problems of geography [2–10]. Basins are hierarchically structured integral formations anchoring many types of cycle of matter, which are isolated within clear-cut orographic boundaries [6]. Transportation of material within basins is carried out under the influence of water flows, which belong to a single erosion-bed system [11], and the basins themselves are complex cascade systems [12]. Examination of the functioning of basins (studies into the redistribution of water flow, sediment load, and dissolved solids) serves to investigate the interaction mechanisms of processes, assess the rate and directedness of transformation of topography, as well as to make short-term forecasts of the evolution of erosion-accumulation processes. Hence, currently, there is a worldwide expansion of geomorphological research into small- and medium-sized river basins in which changes of the influence of separate factors on the regime of functioning of lithodynamical systems are manifested relatively rapidly. Of particular relevance are such investigations for basins in which the climatic fluctuations were accompanied by dramatic changes in land use practices [13]. Significant rearrangements in the dynamics of erosion-accumulation processes should be expected in agricultural regions of the south of Siberia, which, since the late 1990s, have shown a reduction in the area of croplands.

The priority entities for understanding the functioning of lithodynamical systems are represented by the river basins of forest-steppes in the south of Siberia, which are distinguished by a high sensitivity and fast response to climate change [14,15]. For such an analysis, we decided to use the basins of the Angara, Selenga, Upper Lena, and Upper Amur, which are adjacent to each other and stretch from west to east for almost 15,000 km (Figure 1).

They are characterized by a pronounced originality of the conditions of functioning. This peculiarity consists of (1) the intracontinental position of the basins, (2) combination of the mountain and plain terrain, (3) alternation of tectonically stable sections and the Baikal rift zone, (4) location within the zone of active influence of the Asian anticyclone, (5) location at the southern border of the permafrost zone and along the northern limits of the vast arid region of Central Asia, and (6) a high degree of agricultural development. The eastern part of the basin is periodically strongly influenced by monsoon activity. The basins are characterized by an uneven course of processes in time; they are periodically of an extreme nature.

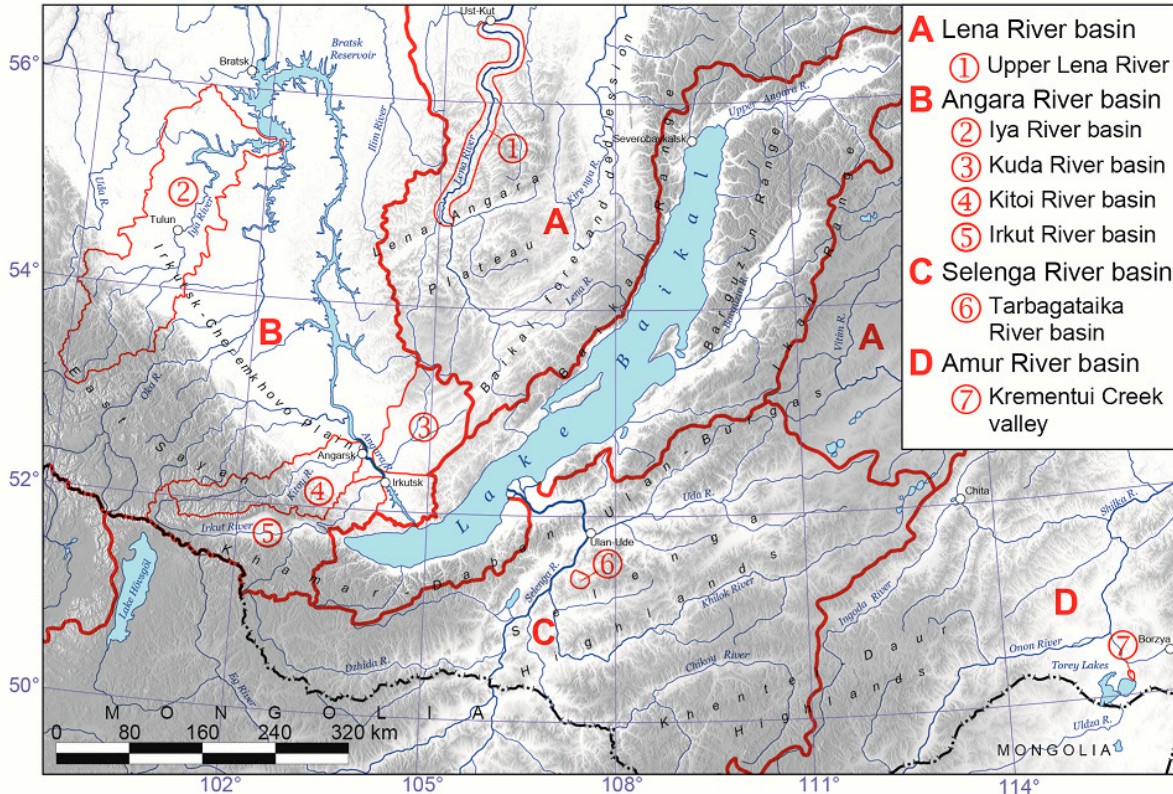

**Figure 1.** Location map of the basins of the Upper Lena (**A**), Angara (**B**), Selenga (**C**), and Upper Amur (**D**). The letters in the circles show the sub-basins of detailed studies.

The basins under consideration belong to various degrees in agricultural production. The period of intensive agricultural development in the south of Eastern Siberia is about 130 years. It is associated with the construction of the Trans-Siberian Railway, then the cultivated area in the Irkutsk governorate increased by 3.5 times. Plowing slope territories caused widespread development of soil erosion and hillwash. Catastrophic manifestation of these processes forced peasants to transfer the arable land to fallow for many years. In the Baikal region, up to 15% of the arable land was abandoned due to soil loss in the 19th century.

The postwar years and the period of mass development of virgin and fallow lands (since 1954) gave rise to the development of erosion processes. Currently, the natural land cover has been changed to 70–90% of the territory in the south of Siberia. As a result of plowing the slopes and pasture

digression, it significantly lost its protective anti-erosion properties, as a result of which accelerated erosion was widespread when the ablation is not compensated by the soil formation process.

Currently, due to the forced conservation of agricultural land, noted since the 1990s of the last century, soil loss from erosion has significantly decreased [16].

Agricultural areas are geomorphologically represented by denudation piedmont and intermountain plains going around the South Siberian mountain belt from the north or shooting out into it occasionally. Here, due to the difficult terrain, about 50% of the croplands are located on slopes with a steepness of more than 3°. Long slopes with a steepness of 5–7° or more are often plowed open. A wide plowing of slopes in Eastern Siberia began during the massive resettlement of peasants from the European part of Russia. The period of intensive agricultural development is about 70–100 years. Plowing of slopes has led to the fact that now more than 2.5 million ha are subject to various degrees of erosion and deflation [17] (Table 1).

**Table 1.** Distribution of eroded and deflated soils in some regions of Siberia, thousand ha [18].

| Region, Republic | Croplands | Soils | | | Total Degraded Soils | |
|---|---|---|---|---|---|---|
| | | Eroded | Deflated | Eroded and Deflated | thou ha | % from Croplands |
| Irkutskaya | 1726.0 | 238.1 | 67.0 | 247.2 | 552.3 | 32 |
| Buryatia | 1018.0 | 272.0 | 308.0 | 250.0 | 830.0 | 81 |
| Zabaikalskii | 2256.0 | 185.6 | 1024.4 | - | 1210.0 | 54 |
| Total | 5000.0 | 695.7 | 1399.4 | 497.2 | 2592.3 | 51 |

It seems interesting and advisable to consider erosion-channel systems in one work to show all possible directions and volumes of matter transfer in the river basins of the south of Eastern Siberia, given the strong economic development of the territory, and the diversity and high spatio-temporal variability of the processes involved in the functioning of these systems. In the presented paper, attention is drawn to the main mechanisms of the functioning of systems, including soil erosion on sloping watersheds, gully erosion, channel deformations, debris flows, and catastrophic floods. The authors suppose such a review will allow not only to show the specific regional features of the fluvial systems of the territory under consideration but also to assess the extent of the environmental threat under various scenarios of the development of processes, and outline ways to solve problems to minimize damage during extreme manifestations of the processes.

## 2. Materials and Methods

Sediment redistribution within the basin was investigated from the system's perspective. An analysis was made of the cascade system whose integrity is determined by unidirectional flows of matter moving by gravity from the upper to the lower hypsometric levels of relief. The individual geomorphological elements are represented by the slopes of interfluves, and the gully-balka and river network [19], which comprise different lithodynamical zones [5]. The rivers' channels are an integrating element of the system. The study of each subsystem and each time interval of the functioning of the basin uses a particular set of modern methods adopted by the international community of geomorphologists [18,20,21]. Additionally, use is made of an integrated technique of studying geomorphological systems in order to carry out a comprehensive analysis of their functioning. Such an approach combines ground-based investigations with remote sensing (by widely using Geographic Information System technologies), the study of soft sediments, mapping of processes, analysis of time series of hydroclimatic indicators of functioning of the basin, and with the use of computational models. Geomorphological work within the basin characterizes the volume of mobilization of matter from slopes and in valley bottoms. The long-term observations of soil washout were part of complex physical and geographical experimental research on field experimental stations

of the V.B. Sochava Institute of Geography SB RAS [22]. These included washout observations using catchment sites, metal benchmarks, and Campbell frames. Surface discharge was also studied in the bottoms of temporary streams. In addition, we performed short-term observations of slope erosion in other regions of Eastern Siberia lasting 3–5 years. The volume of washed-out soils on croplands was determined by the Sobolev's method [23]. To assess the intensity of basin erosion, we used time series of sediment runoff over 25 basins of southern Siberia. To determine the potential flush, we used a technique developed in the research laboratory of soil erosion and channel processes at Moscow State University [24]. Its applicability in revealing the average annual rates of flushing in the southern regions of Eastern Siberia is confirmed by the data of field measurements of the rate of diluvial processes [25]. The correlation coefficient of measured and calculated rates is quite high (0.86 +/− 0.11). The methodology is based on the universal equation of soil loss as a result of storm runoff [26]. Maps are compiled on the main agricultural enclaves of the south of Eastern Siberia [27]. To quantify the matter redistribution in the upper links of the erosion network, we used empirical models of melt and storm runoff. We have previously checked the possibility of using the models for forest-steppe regions of Siberia by the data of field experimental studies [25]. As a result of the calculations, maps of the distribution of zones with different intensities of storm runoff were compiled.

For integration with GIS technologies and for quantitative assessment of soil erosion in the Tarbagataika basin (Selenga Highlands), we used the RUSLE (revised universal soil loss equation) model, which takes into account four main factors affecting erosion: Sediment erosion, soil erosion, topographic factors, and land cover and land use [28]. The satellite images of the modern erosion network from Google Earth were interpreted to identify the distribution areas, morphometry, and morphology of erosive landforms. The spatial and temporal dynamics of gully forms were studied using tacheometric surveying and permanent benchmarks.

The assessment of the plane channel deformations of the Lena river over a period of more than 100 years, used *MapInfo* software and compared the navigation maps of the Lena river published in 1912 [29] and modern large-scale topographic maps, and aerial imagery (surveys made in the 1980s of the past centuries) and modern space images from various resources. We used retrospective topographic maps of a scale of 1:84,000 published in 1896–1914 [27], topographic multidate maps of a scale of 1:100,000, and Earth remote sensing data (Landsat; 1976–2015) for the analysis of morphology and plane deformations of the Irkut river channel [30].

The assessment of the current state of exogenous geological processes (EGPs) within the river valleys used a perspective survey from unmanned aerial vehicles DJI PHANTOM 3 and 4.

## 3. Results and Discussions

### 3.1. Suspended Sediment Yield

The sediment yield is an integral indicator of the intensity of fluvial processes in all parts of the erosion-channel system of the river basin. The modulus of suspended sediment yield includes the basin and channel components. Its long-term dynamics reflect the influence of economic activity and climate change on the sediment redistribution within the basin [31]. Hence, this indicator is important in assessing the functioning of basin systems, and their contribution to the total land denudation [32].

Analysis of long-term runoff series of suspended sediment lasting from 20 to 44 years revealed significant differences between basins in the rate of fluvial processes (Figure 2).

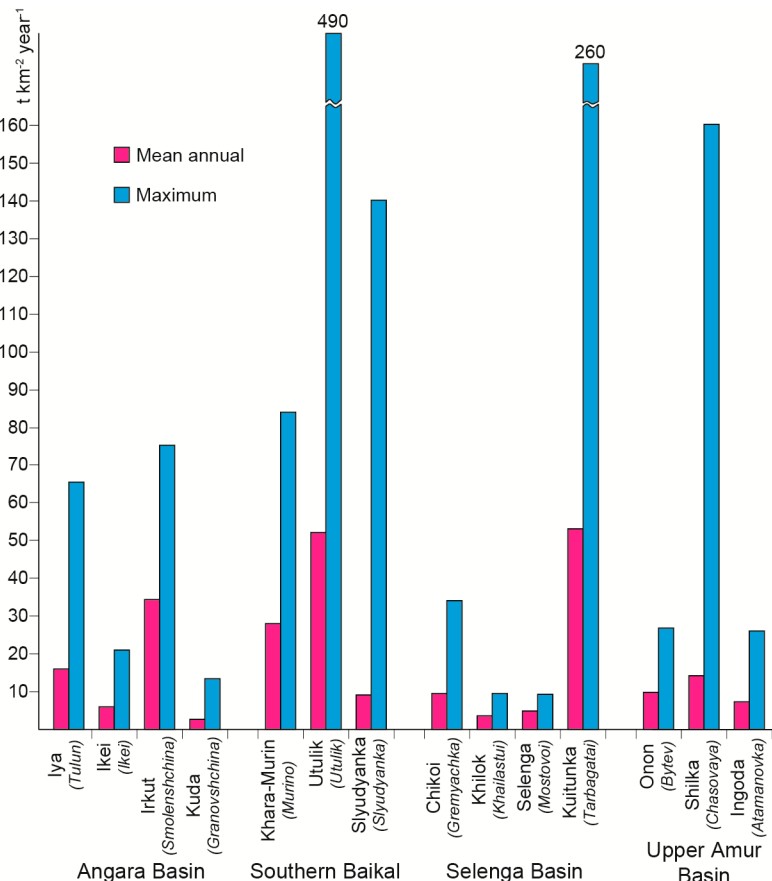

**Figure 2.** The ratio of the mean annual and maximum modulus of suspended sediment yield in the basins of Angara.

The highest values of sediment yield were recorded on the rivers flowing from the Khamar-Daban ridge into Southern Baikal (see Figure 2). The mean long-term modulus of suspended sediment yield in the whole region is 30 t/km$^2$ per year. Maximum values for the observation period reach 490 t/km$^2$ per year (the Utulik river), and they are associated with mudflows and characterize the basins of these rivers with extremely high activity. The Angara basins occupy the second place in the intensity of functioning of erosion-channel systems; their rivers (Figure 2) originate in the mountains of the East Sayan. The mean modules of suspended sediment yield are 15 t/km$^2$ per year; the mean maximum modules in the region are 44 t/km$^2$, and the maximum at the Iya and Irkut rivers reach 65.3 and 75.3 t/km$^2$ per year, respectively. These rivers are characterized by rare catastrophic floods with a high risk for human life and high rates of channel deformations.

The basins of Transbaikalia (Selenga and Upper Amur) are characterized by approximately identical distribution of moduli of suspended sediment yield. Basins with extremely high sediment yield stand out against the background of relatively low values of this indicator (5–10 t/km$^2$ per year), typical of most rivers (see Figure 2). Within the Selenga basin, it is the Kuitunka basin (see Figure 2), where mudflows, gully erosion, and intensive flushing from croplands are periodically observed [33,34]. In the upper part of the Amur basin, the Shilka basin stands out (Figure 2), which is characterized by periodically recurring extreme geomorphological events associated with monsoon rains, manifested in the activation of erosion and channel processes that move a large amount of sediment in the basin.

The basins under consideration are located in the cryolithozone; their functioning occurs under the intensive development of ground heaving processes, frost cracking of soil, and aufeis formation [35,36]. Cryogenesis also affects the dissolved solid yield [37]. It was established that in the mountainous regions of the south of Eastern Siberia, the annual values of dissolved solid yield are several times

higher than the values of solid yield [37]. Our studies also showed that in the Kuda basin (Central Siberian Plateau), the ionic flow is 7 times higher than the suspended sediment yield [38].

The analysis of the long-term dynamics of the suspended sediment yield revealed its clear cyclical course, due to climatic fluctuations [36,39]. This indicator is also very sensitive to the changes in economic activity in the basin. So, the conservation of agricultural land within the Selenga basin in the late 1990s caused a downward trend in the suspended sediment yield [17,40]. Consideration is given to the redistribution of sediment in various parts of the basin systems.

### 3.2. Hillwash

The rainfall regime is the decisive factor for the occurrence of slope runoff and wash in the south of Eastern Siberia. Here, the main relief-forming role is played by storm rainfalls of the second half of summer. Additionally, the wash intensity depends on the erosive ability of storm rainfalls, which varies in individual regions of Siberia [27]. Granulometric analysis of the washed-out fine earth showed that the erosive ability of storm water is higher than that of melted snow, since larger material is washed out during storm rainfalls. A fraction of coarse dust, which has the least resistance to washing, is carried out more easily from the slopes [27,36]. If in the Angara basin about 30% of the annual volume of solid yield is associated with melt runoff, then in the more eastern basins its role is close to zero, since snow most often evaporates during melting.

In total, 2 (1972) to 18 days (1975) with runoff are observed in the steppes of the Upper Amur basin, according to catchment sites on the northern and southern steppe slopes with a steepness of 3–5°. The runoff coefficient in some years varies from 0.10% to 0.37% of the amount of summer rainfall, amounting to 140 to 270 mm in different years. At the same time, the wash averages 0.5–0.7 tha$^{-1}$ on slopes with a naturally sparse grass stand, the amount of washed material increases by 1–2 orders on croplands, and the amount of wash from arable slopes can reach 240 tha$^{-1}$ per year in years with extreme rainfall. In this case, a dense network of channel erosion is formed up to 30 cm deep and 150–200 m long, and fresh outflow cones are formed at the foot of the slopes [41].

In total, 20–70 tha$^{-1}$ of soil at slopes of 4–6° is washed within the Angara basin, depending on the nature of soil cultivation, and 70–200 tha$^{-1}$ on slopes steeper than 8° [42]. Plowing along the relief contours reduces the wash intensity 3–4 times. Most of the washed-out material accumulates at the foot of slopes (from 0.3–2.6 mm/year on deluvial plumes to 55 mm/year below the channel erosion on cropland), in numerous ponds (about 10% of sediment. Distinct 3–5 summer cycles of their course are revealed. The correlation of the zones of wash, transit, and accumulation of matter with the morphological elements of the slopes was revealed [38].

In agricultural areas, zones with potentially high wash will be larger in basins with complex terrain. Table 2 shows the basins of the Tarbagataika (Selenginskoe Highlands) and Irkut (Baikal Rift).

**Table 2.** Distribution of zones with different intensities of potential soil loss in river basins in the south of Eastern Siberia, %

| Soil Loss Intensity, t/ha Per Year | The Basin of the Tarbagataika (Selenginskoe Highlands) | The Basin of the Irkut (Tunkinskaya Depression) | The Dasin of the Unga (Irkutsk-Chremkhovo Plain) | The Dasin of the Kuda (Central Siberian Plateau) |
|---|---|---|---|---|
| ≤2 | 55 | 30 | 19 | 60 |
| 2–5 | 9 | 25 | 21 | 19 |
| 5–10 | 11 | 25 | 45 | 15 |
| >10 | 25 | 20 | 15 | 6 |

The impact of other factors on soil erosion will be considered on the example of the Kuda (area 8040 km$^2$) and Tarbagataika (210.4 km$^2$) basins, and their position is shown in Figure 1. One of the main factors of the development of hillwash is represented by the erosion index of storm rainfall (R$_{30}$)

determined for a 30-min interval. Its mean annual values within the basin are 5–7, but intermittently, they exceed 20 and characterize the basin as an area with a serious erosion hazard of storm rainfall.

The modulus of suspended sediment yield is an important indicator of the functioning of the basin. An analysis of its dynamics in the outlet section at the village of Granovshchina for the 75-year period showed a significant impact of agricultural activity on soil erosion [38]. The entire series clearly shows two peaks of maximum suspended sediment yield corresponding to the beginning of the Virgin Lands Campaign (1962–1975) in this region, and to the period of the highest degree of plowing of lands (1984–1987). A quantitative assessment of the total wash from the melt and storm water runoff allowed for erosion zoning of the basin (Figure 3). It shows that particularly high indicators of storm rainfall wash are observed in burns and in areas of intensive felling. Almost everywhere the rates of melt water wash are minimal: In 85% of the area of the basin, wash does not exceed 0.5 t ha$^{-1}$ per year $^{-1}$.

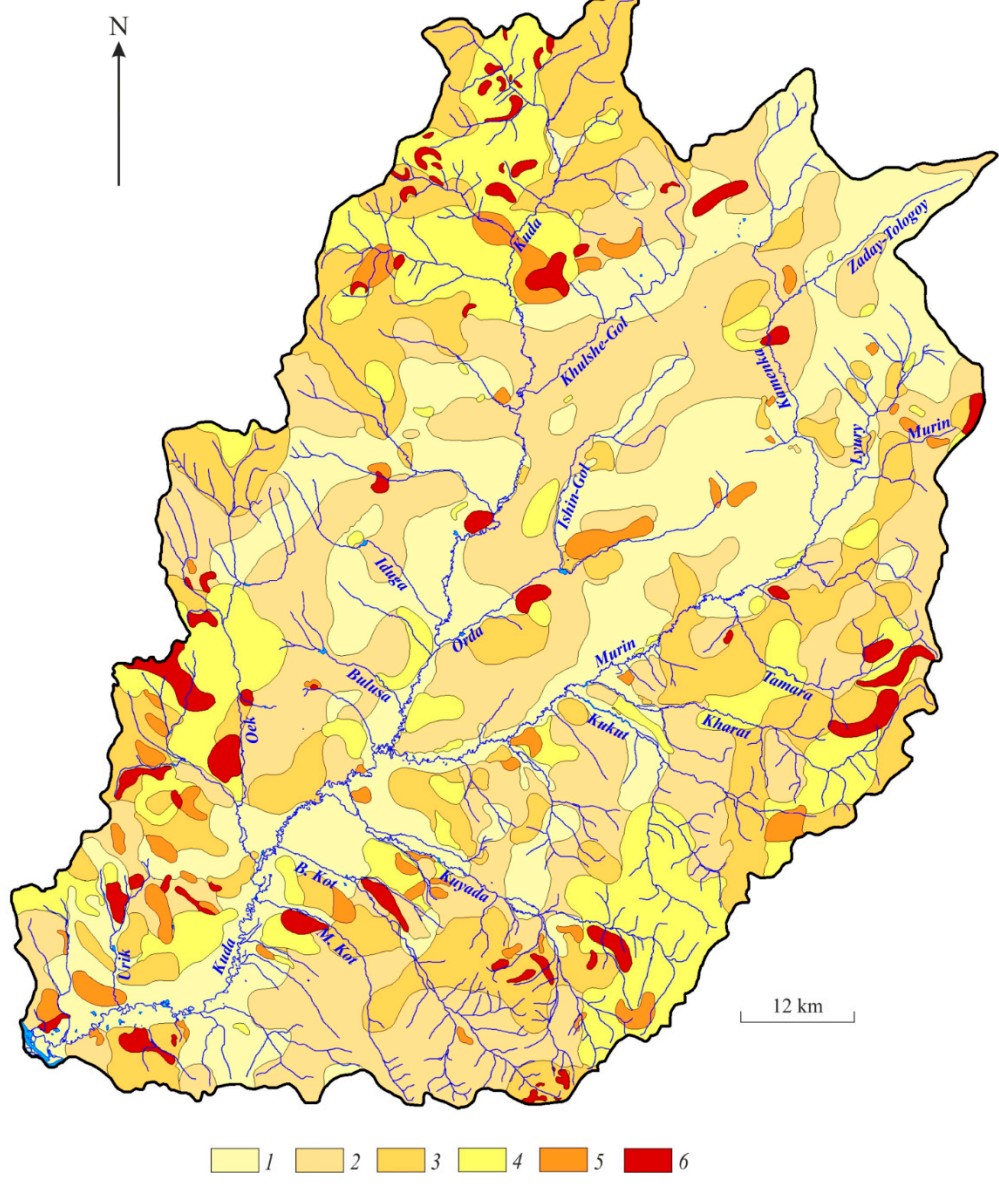

**Figure 3.** Erosion zoning of the Kuda basin. Zones: 1. no erosion hazard (loss less than 1 t ha per year); 2. Weak erosion hazard (loss 1–2.5 t ha); 3. Moderate erosion hazard (2.5–5 t ha); 4. Medium erosion hazard (loss 5–10 t ha); 5. Strong erosion hazard (10–20 t ha); 6. Extreme erosion hazard (loss more than 20 t ha per year).

The Tarbagataika basin (the left tributary of the Kuitunka river) is characterized by a high erosion potential of the relief, typical of the Selenga highlands. The values of potential soil loss, calculated according to the RUSLE model, are extremely unevenly distributed across the basin. They reach the highest values on cultivated lands (croplands and fallow lands) located on slopes (up to 113.38 t/ha$^{-1}$ per year$^{-1}$), with average values for the whole basin of 7 t/ ha$^{-1}$ per year$^{-1}$; the smallest values correspond to forested areas (not more than 1 t/ha$^{-1}$per year$^{-1}$) (Table 3) (Figure 4).

**Table 3.** Soil loss and land cover classes in the Tarbagataika basin.

| Land Cover | Area, % | Area, ha | Range of Soil Loss, tha$^{-1}$·year$^{-1}$ | Mean of Soil Loss, tha$^{-1}$·year$^{-1}$ |
|---|---|---|---|---|
| Cultivated lands | 29.0 | 6098 | 0–113.38 | 19.24 |
| Forest | 55.5 | 11679 | 0–0.85 | 0.26 |
| Grassland | 14.5 | 3046 | 0–28.19 | 5.54 |
| Artificial surfaces | 1.0 | 216 | 0.12–23.55 | 3.15 |
| All classes | 100 | 21040 | 0–113.38 | 7.03 |

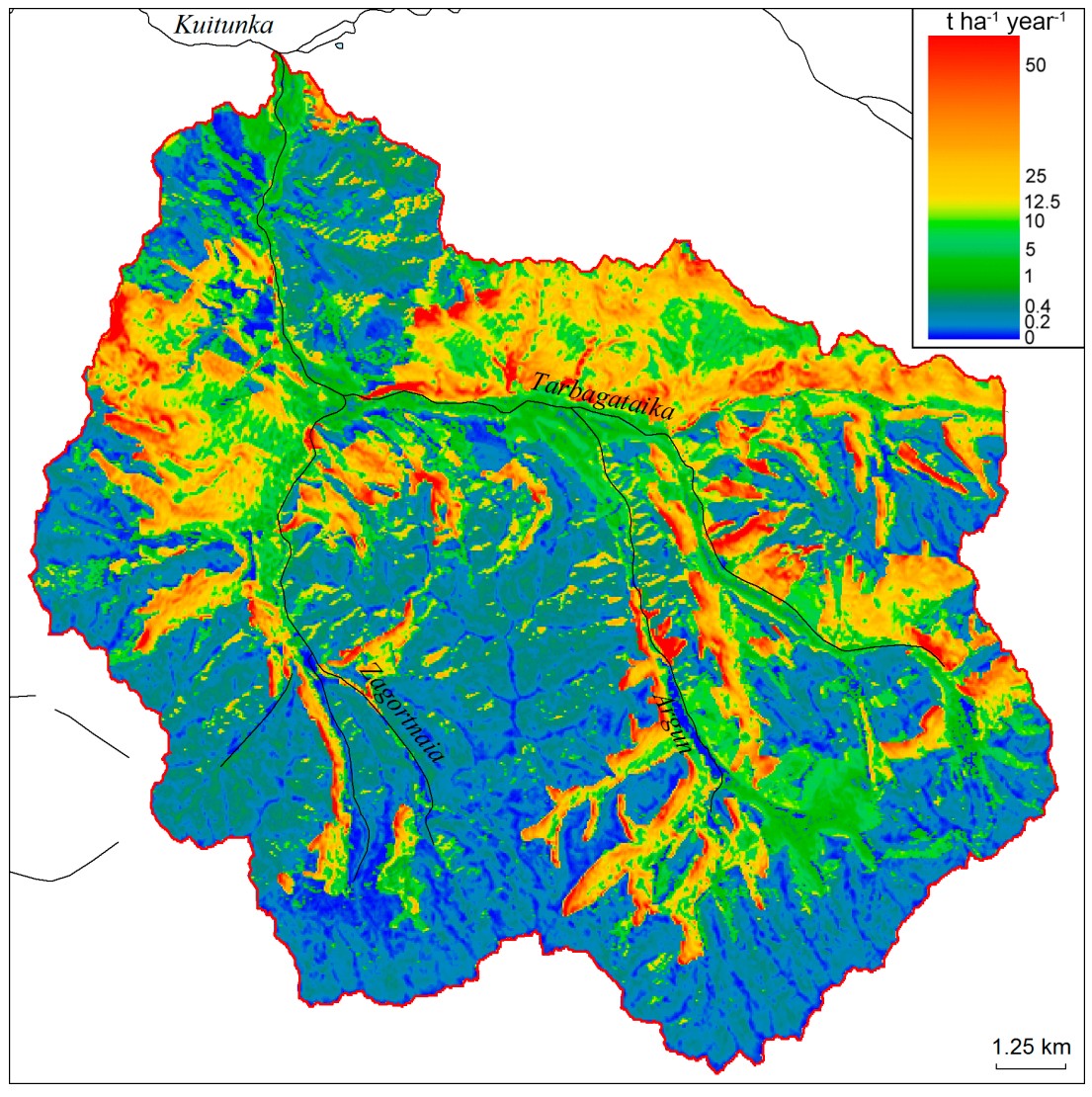

**Figure 4.** Soil loss in the Tarbagataika basin predicted (tha$^{-1}$year$^{-1}$).

### 3.3. Gully Erosion

Gully erosion impacts the mobilization, transport, and resedimentation of matter within the river basins. High development of steppe islands and forest-steppes within the upper Angara and especially small rivers contributes to a complex structure of erosion-accumulative systems, represented by erosion belts. O. P. Ermolaev reports [43] that on the slopes of river basins, the following erosion belts are distinguished: Rainsplash destruction, micro-rill-washing, rill erosion, gulling, predominant accumulation, and erosion-free areas. In southern taiga, the erosion-free belt covers significant areas. We developed an erosion belt scheme for the area with severe gulling processes in the Upper Angara basin (Figure 5). The micro-rill erosion belt is found on the plowed sides of the basins, especially after heavy showers, with a strip 50–100 m to 1200–1500 m. Within this belt, micro-rills are saturated with suspended matter and resedimented within depressions on the slopes or on the floodplain at the foot of the side's valleys. The morphometry of the channels of micro-rills is characterized by a small incision depth of up to 3–5 cm, a width of 10–20 cm, and a length of several meters to tens and sometimes hundreds of meters. The rill erosion belt with a width of 100–300 m to 1500–2000 m occupies a significant area. The rill net on more gentle slopes, on the steep sides of the valleys of straight gullies, is manifested in the relief.

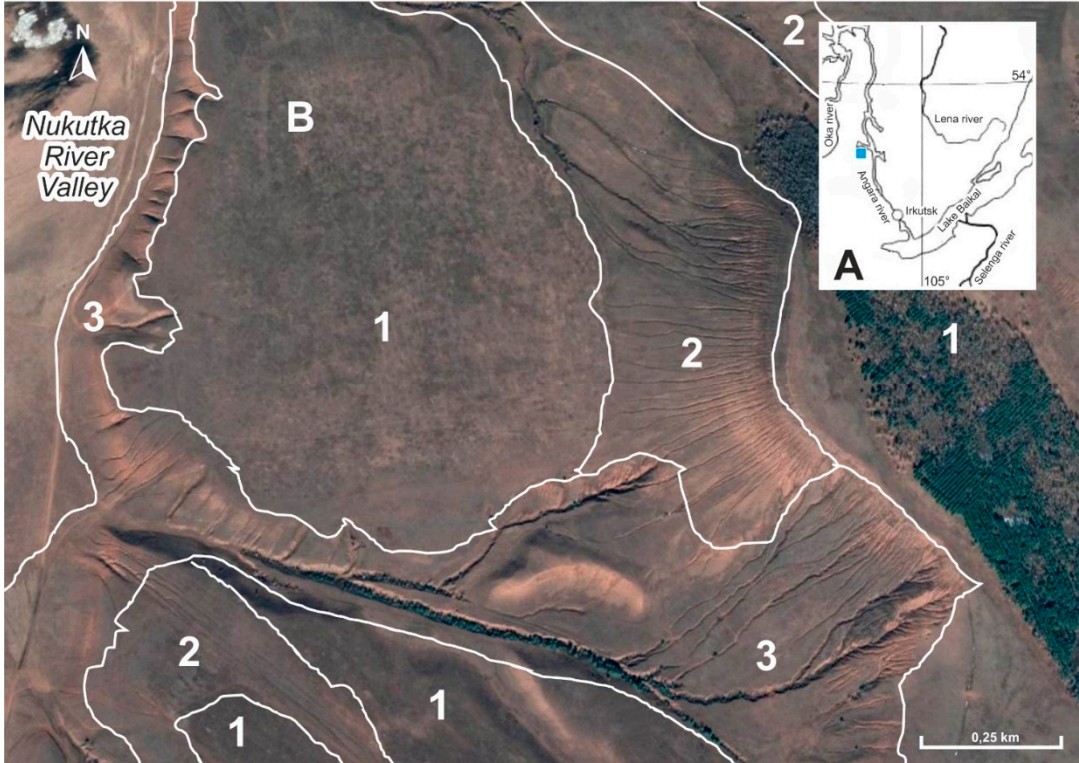

**Figure 5.** Erosion belts scheme for the area, adjusting to the Unga Bay of the Bratsk Reservoir. (**A**). Location of the area (marked with a blue square), (**B**). Erosion belts: 1. splash erosion, 2. Rill erosion, 3. Gully erosion. Based on the satellite image Google Earth. Coordinates: 53°43′37.16″ N; 102°49′33.56″ E. The image was taken on 4 October 2019.

Sometimes, they are found as a continuous strip. The rills are located along the entire length next to each other; their depth is 30–50 cm, width is from 50 cm to 1–3 m, and the length depends on the length of the slope. The gully erosion belt occurs locally and discretely, often along one steep side of the valley.

Large valley-balka systems are formed on the right bank of Unga Bay. Their length is 5–10 km and the valley cutting is 2.5–3.5 m. They are formed in red-colored terrigenous-carbonate rocks of the Cambrian and terrigenous-carboniferous rocks of the Jurassic in marls, mudstones, and calcareous

sandstones with intercalations of gypsum. The deluvial-proluvial sediments are represented by sandy loam and loam. The transverse profile of the valleys is asymmetric, the right wall is steep, and the left one is gentle. Slope rills cut the walls and fresh bottom rills are observed along the channel.

This belt is well defined near the coastal gullies. Its width is 50–100 m and more. Moreover, individual gullies go beyond the valley and reach several kilometers in length and have a dendritic shape. The upper boundary of this belt is often the boundary of cropland. Thus, the maximum flow of soft sediments into the floodplain and, accordingly, into the river bed occurs along the gully erosion belts. The maximum is observed in July, when storm runoff is formed in the channels of gullies. So, the mean suspended sediment discharge near the village of Balagansk (the Angara basin, the upper part of the Bratsk reservoir) ranges from 30–40 kg/s, and then after heavy rains it reaches 700 kg/s and more.

A large number of erosion forms per unit area (25–50 units/100 km$^2$) deeply cut with a V-shaped transverse profile were recorded on the ledges of the high basement terraces of the Lena river near the settlement of Kachug (Figure 6).

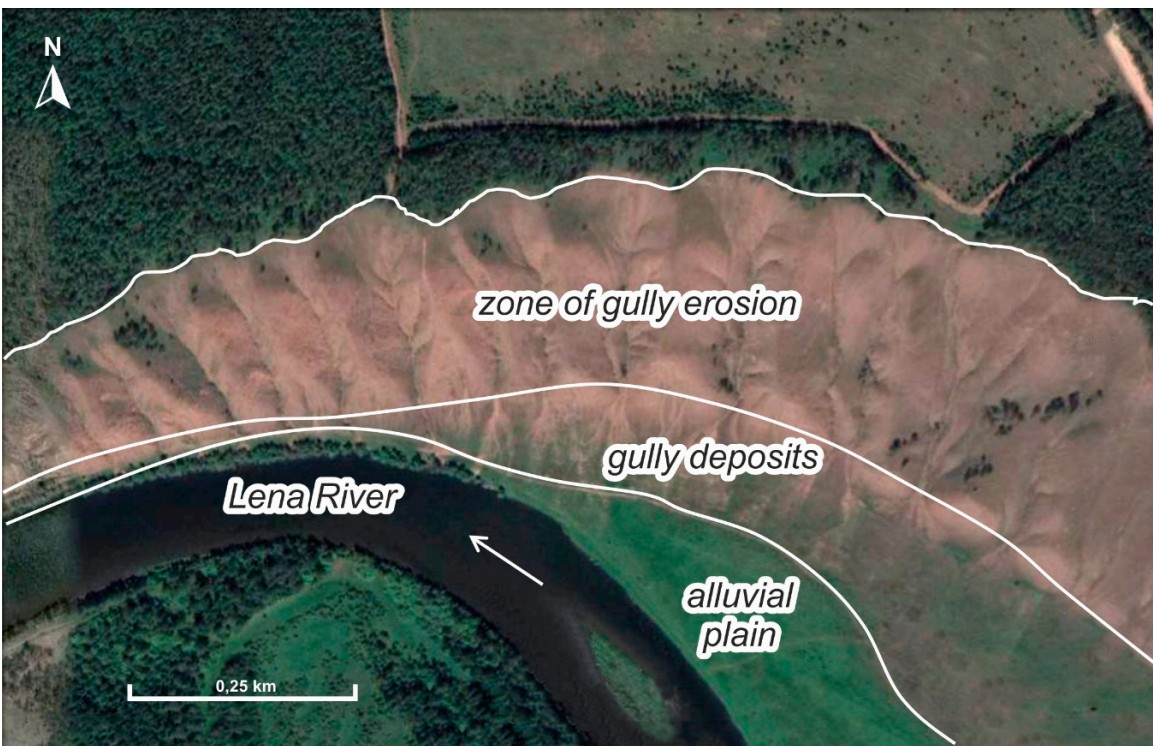

**Figure 6.** Gullies cutting the steep walls of the Lena river valley near the settlement of Kachug. Coordinates: 53°59′42.93″ N, 105°49′48.40″ E. The image was taken on 19 July 2018. Based on the satellite image Google Earth.

In general, the Angara region is characterized by the maximum density of erosional forms of 10–25 units/100 km$^2$. Small basins on the gentle sides of the valleys, such as the Kuda basin, are characterized by weak gully dismemberment. It is dominated by short (as short as 100–200 m) rills, and by gullies 1–3 m in depth. Characteristic for the near-watershed gullies is their growth in the estuarine part. Their heads often abut against escarps up to 50 m in height, which are composed of Cambrian sandstone. Over the last three decades, a reduction in croplands and a decrease in livestock has given rise to new gullies in areas of felling and the growth rates of their heads do not exceed 0.2–0.5 m/year.

In the basins of the Selenga, Onon, and Argun, gullies are widespread in areas with a high degree of anthropogenic load, often in settlements. Here, their density reaches 50 units/100 km$^2$ or more.

The density of gullies in the catchment area of the Kuitunka river (Selenginskoe Highlands) in thick loesslike sandy loams, for example, is 390 units/km$^2$. In forest-steppes, gullies reach their

maximum proportions (in loess-like sediments, their length is up to 4–6 km, depth: 20–30 m; in sandy sediments, their length is up to 1 km, depth does not exceed 3–5 m). On the terrace of the Onon river between the villages of Ikaral and Ust'-Borzya, there are 16 large linear gullies with a length of up to 800 m and a depth of 5 m (Figure 7).

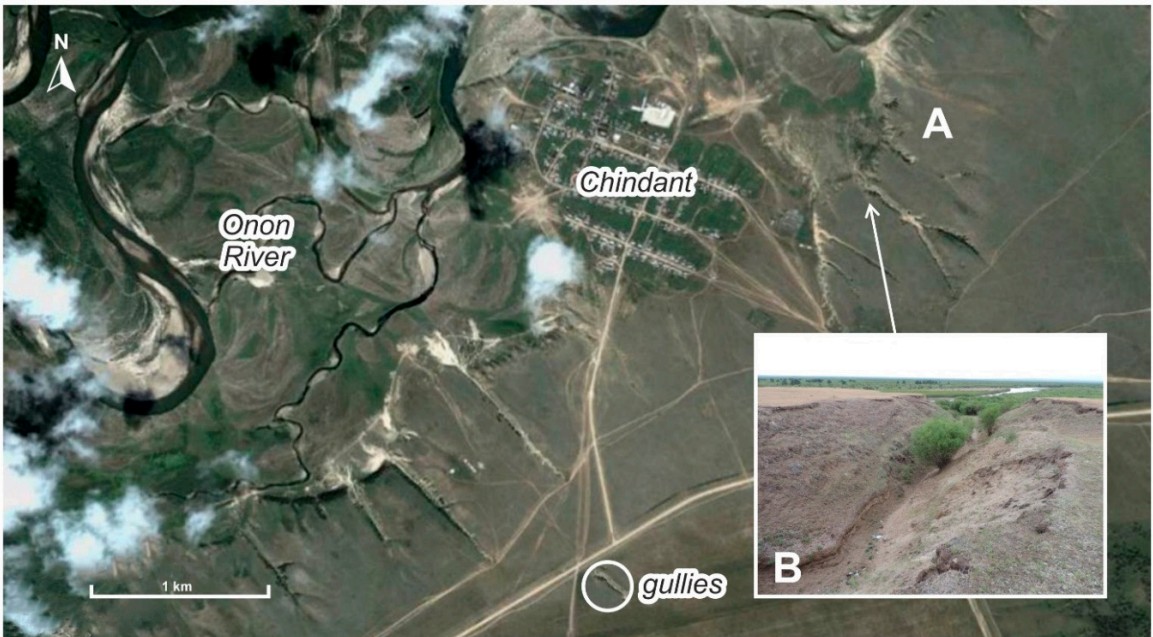

**Figure 7.** Stretch of the Onon river valley. Coordinates: 50033′44.29″ N; 115023′32.13″ E. The terrace cut with gullies (**A**). The image was taken on 8 March 2019. (**B**) Image of the gully (taken on 16 August 2015), its location is shown with the arrow.

Slope gullies dismember the high walls of lake basins (Lake Zun-Torei, Lake Chindant, Lake Gusinoe, etc.). The mean density of the gully net is 0.2–0.5 km/km$^2$. Bottom erosion of an oval shape with gully tributaries with a length of 20–40 m, a width of 6–8 m, and a depth of 1.5–3 m. Sloping (coastal) gullies are most often linear in shape with a V-shaped transverse profile, 5–10 m wide, 1–3 m deep, and in some cases up to 8 m, with a length from 200–600 m to several kilometers.

Cropland is actively eroded on slopes of low mountain ranges in the steppes of southeast Transbaikalia (the Upper Amur basin). Local and areal linear forms are found here, which are confined to natural and artificial (arable furrows, road network, etc.) drainage lines. Waterways from the fields are formed during precipitation, with a total amount of more than 5 mm and an intensity of 0.02–0.03 mm/min. Stormflows during rains with a total precipitation of 10–20 mm and an intensity of 0.10–0.50 mm/min in places with natural depressions break through the furrows and flow down the slopes. They give rise to local micro-rills and rills with a stepped longitudinal profile. Of particular hazard are storm rainfalls with a total precipitation of more than 50 mm and an intensity of more than 1.0–1.5 mm/min, as the fields are covered by a network of rill erosion. After the storm rains on 10–13 July 1979, for example, with a total precipitation of 54 mm and a maximum intensity of 1.8 mm/min on a plowed slope with a steepness of 3–5° northwest of the village of Kharanor, the density of local erosion had a length of 150–200 m, and a depth of 30 cm was 10 units/km$^2$. Moreover, the entire surface of the cropland was subject to a continuous sheet and fine-rill erosion. In some areas, the upper soil horizon was washed along with sowings; the height of the wheat germ reached 20 cm.

The removal of fine earth from the field amounted to 240 m$^3$/ha, on average. In the Kunaleika river basin (Western Transbaikalia), after heavy rains in the summer of 1988, 42 new gullies 12 km long were formed, 37,000 t of unproductive soils were resedimented, and 28 ha of cropland were removed from use [44].

The results of the multi-year study of gully erosion in southern Siberia indicate a very uneven development of erosion forms over time. The linear forms of erosion are characterized by a stormy initial stage of development associated with a shower and steady rain. Cropland and especially fallow land on slopes with a steepness of 3–10° during shower and during snowmelt are subject to severe and very strong erosion. In ordinary moderately humid years, the rates of washout and erosion are substantially lower. The dynamics of gully formation has a pulsating character, when after several years of a total absence of gully head growth, more active growth (1–3 m/year or more) is observed in humid and extremely humid years. It was found that most forms of erosion have reduced the rate of regressive erosion to 0.3 m/year over the past 15–20 years. The decrease in the activity of gully formation is caused by a decrease in the area of plowed land. Data on growth rates were obtained by monitoring the growth of more than 150 peaks of gullies in different regions of southern Siberia at physical and geographical field stations (26) as confirmed by the data of other researchers [34,45]

### 3.4. Channel Processes

Channel processes in these basins vary widely. Natural processes, including the aeolian migration of matter, have an active influence on their course [46]. To determine the influence degree of the geological and geomorphological structure of the territory, hydrological conditions, and natural processes of channel development on dynamics of channel deformations, we analyzed the morphological changes of channels of the Lena river (upper course and flat-platform part) and Irkut (Tunkinskaya depression, Baikal rift zone, and the Irkutsk-Cheremkhovo plain of the Siberian platform in the lower reaches) over a 100-year period.

The entrenched channel type is characteristic of the Lena river valley section located within the Lena-Angara uplift of the Irkutsk amphitheater [47] at the regional level. We analyzed changes in the morphological elements of the channel from the mouth of the Tutura river to the mouth of Turuka over a 100-year period and revealed three sections with different erosion-accumulative components. The area with directed accumulation coincides with the Typto-Tuturskii declivity, the area with balanced erosion-accumulative processes coincides with the antecedent area of the valley, and the relatively stable area without obvious signs of plane channel deformations with the arcuate bed of the Lena river valley. Stretches where the processes of accumulation and erosion compensate each other occupy the largest part (138 km) of the investigated part of the Lena channel. Further along the distribution scale, there are stretches with a predominance of accumulation processes (130.1 km), almost 20 km of which are local areas with accumulative processes only in the mouths of tributaries. They are followed by unchanged (stable) stretches, occupying 50.9 km, as well as an area with a predominance of erosion processes (occupying 15 km). The estuarine stretches of large tributaries, as well as branched sections of the Lena river channel, naturally underwent the most noticeable plane channel deformations. Thus, within the mouth section of the Tutura river (Figure 8), we registered the death of the floodplain and secondary shallow by-channels, and the formation of new islands with an increase in their total area by 0.25 km$^2$. At the mouth of the Orlinga (Figure 8), on the contrary, there is a predominance of erosion processes over accumulative ones. Here, we recorded a decrease in the number of islands, their erosion, fragmentation, and formation of new channels. The nature of the trend of plane channel deformations within the site under investigation is associated with the natural development of channel forms due to the geological, geomorphological, and hydrological conditions of the territory.

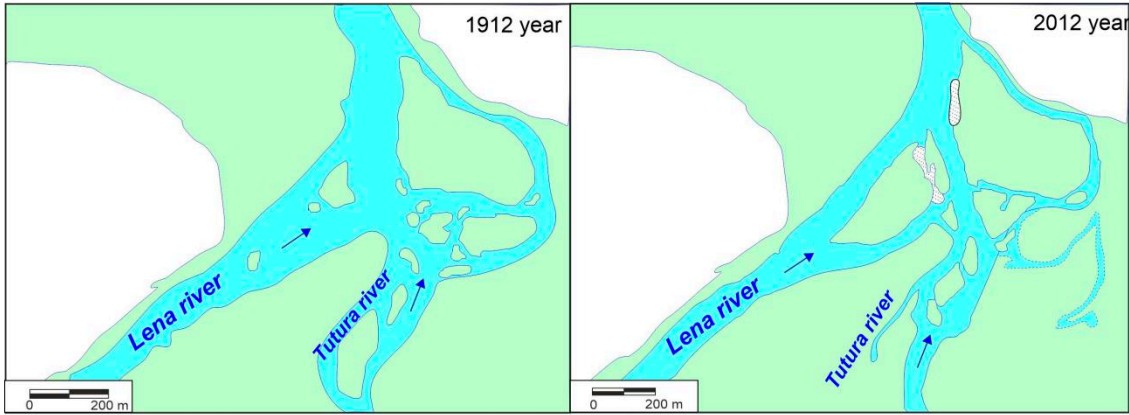

Estuarine stretch of the Tutura river (Typto-Tuturskii stretch of directional accumulation)

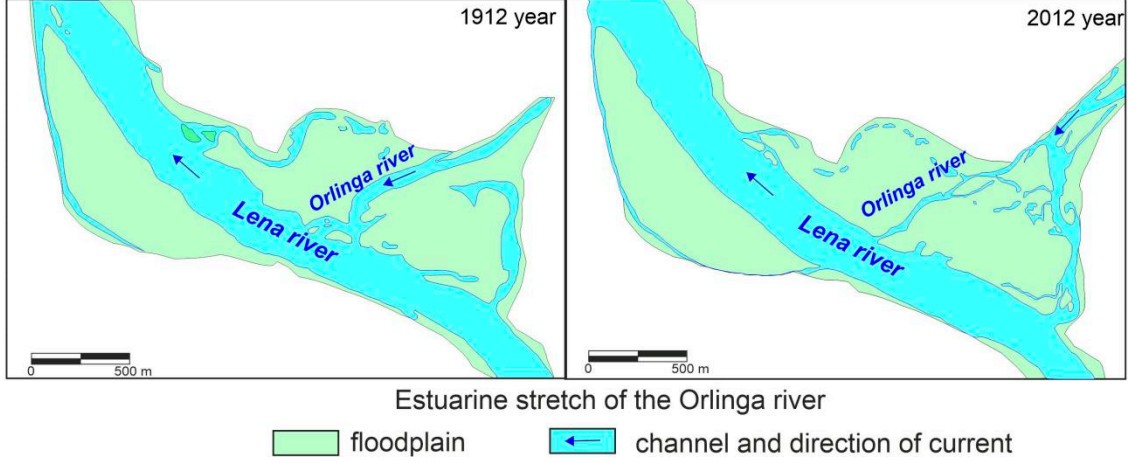

Estuarine stretch of the Orlinga river

floodplain    channel and direction of current

**Figure 8.** Channel plane deformations of the Lena river.

The morphodynamics over 100 years of the Irkut river within the Tunkinskaya depression and the inter-hollow bridges associated with it [31] showed the most intensive development of channel deformations within the basin section, where a broad-floodplain channel type developed.

We revealed 64 bends in total, measured the main morphometric parameters of the bends, and determined their morphodynamic types on the site of free meandering on the stretch of the village of Shimki at the mouth of the Tunka river. The most intense channel deformations were recorded in the zone of young tectonic subsidence (Figure 9). The broad-floodplain stretches of the divided-sinuous type and the area with the development of steep loop-like meanders turned out to be the most stable (Figure 9). The total length of relatively stable stretches was 80 km (60%) of the stretch under investigation.

The analysis of different time images identified the most mobile stretches of the channel within the lowlands of the Irkut river [48]. In two cases, these are floodplain braiding with meanders in the left and right binnacles connected by minor by-channels. Another stretch is the meander in the right binnacle, complicated by the development and dying of by-channels. The main types of horizontal deformations of channels were revealed: In space, these are the forms of local (reformation of meanders) and local (changes within the elements of meanders, dying of transverse by-channels in floodplain braiding, an increase and decrease in the area of islands, caving and accretion of the banks) levels (Figure 10).

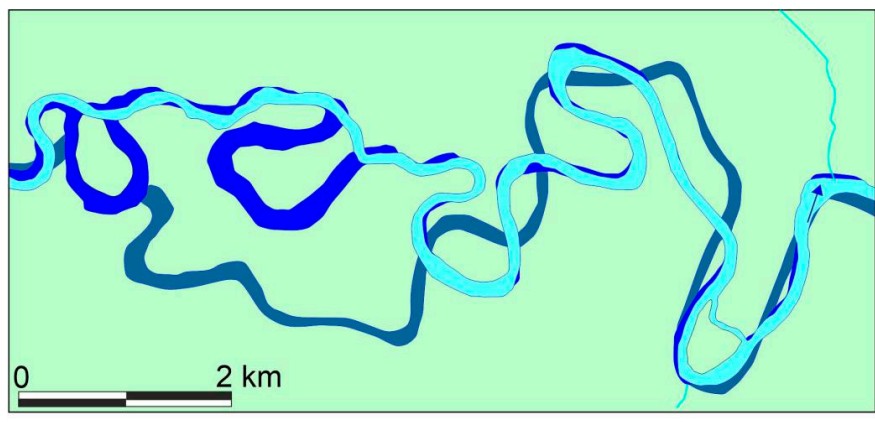

Reorganization of the Irkut channel on the stretch
of Lake Barun-Nuga - the Kyren river

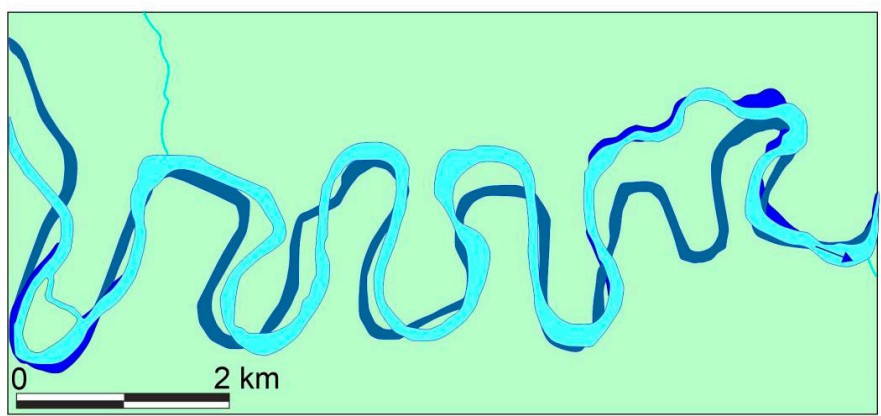

Relatively stable position of steep loop-shaped meanders of the Irkut
on the stretch of the Kyren river - Lake Bandin-Khol

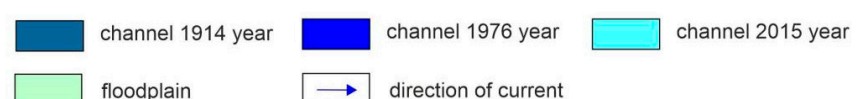

**Figure 9.** Comparison of the channel plans of the Irkut river in three time sections.

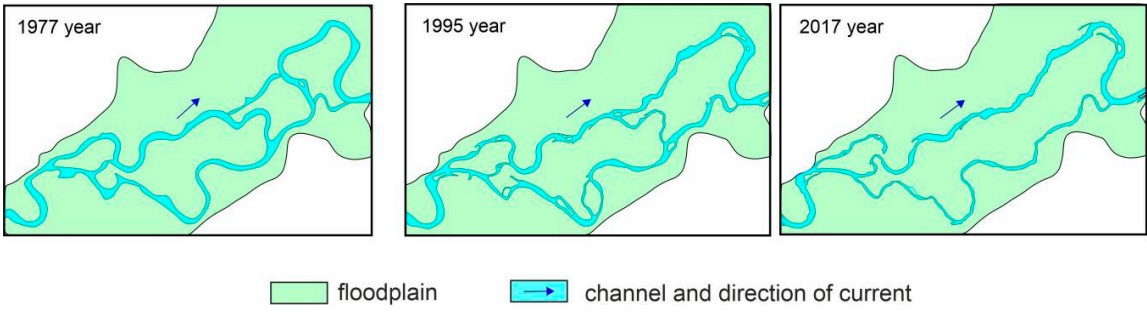

**Figure 10.** The most dynamic stretches of the Irkut channel (Maksimovshchina—the mouth).

Thus, we can register common features of the development of channel processes that were revealed in the analysis of channel deformations located in different geodynamic settings. They consist in the fact that the morphostructural structure of the territory and the influence of the neotectonic regime are the determining factors in the formation of morphodynamic channel types at the regional and local levels, such as for the broad-floodplain type of the Irkut river channel within the Baikal rift zone, and for the incised type of the Lena riverbed within the region of the Lena-Angara uplift of the Siberian platform.

### 3.5. Coastal Deformations of the Rivers of Eastern Siberia

Monitoring revealed a significant variability in the development of slope processes in the same coastal ledges, associated with the water saturation degree of the soil, as well as with the water level in the river and seasonality. We recorded maximum rates of coastal destruction within the plains of up to 1.5 m per year. Such deformations are characteristic of coastal benches composed of sandy-loamy sandy sediments, with the subsidence of soil blocks, with their further caving and shedding. The longest stretches of coastal destruction are located within convex banks at the tops of free bends. We managed to obtain the coast retreat rates during the catastrophic floods of 2019 in these territories for rivers in the mountainous and foothill areas of the left bank of the Angara and the framing of Lake Baikal. For mountainous areas in loamy sandy benches, they amounted to 2 m, and in pebbled areas reached 6 m due to caving processes. Thus, the volumes of material supplied to the channel reach 1800 $m^3$ within a 600-m-long stretch, a bench height of 2 m, and an average retreat of the bench edge of 1.5 m.

The creation of huge reservoirs on the Angara significantly complicates the ecological situation in the basin due to the sharp intensification of the development of landslides, karst, gullies, suffosion, and intensive erosion of the banks of the Irkutsk, Bratsk, and Ust'-Ilim reservoirs, which causes great damage to forestry and agriculture. According to [49], over the period of exploitation at the end of the 20th century, the erosion in loose sediments amounted to 70 m along the Ust'-Ilim and 200 m along the Irkutsk and Bratsk reservoirs. On the banks formed by rock and semi-rock, this value did not exceed 80 m. The length of the abrasion coasts is 51%, 34.3%, and 34% of the total length of the coastline, respectively. The stream-bank erosion zone on the Irkutsk and Bratsk reservoir included residential and utility buildings, and gardening facilities. The total area of lost lands due to the bank erosion on the Bratsk reservoir during its operation is about 45,000 ha, about 4000 ha of which are forests and shrubs, and 420 ha are cropland and household plots. In total, 114 ha of arable land have been eroded in the Irkutsk reservoir, about 210 ha of forests and shrubs, 25 ha of household land, and in the Ust'-Ilimsk reservoir about 600 ha [49]. The deformation of coastal slopes of valley reservoirs has retained a high degree of activity in the 21st century, and periods of karst activity were noted in 2003, 2009, and 2011 [50].

### 3.6. Extremals of Fluvial Processes

In recent years, the number of natural emergencies in Russia and in the world has increased. One of the factors initiating the increase in the frequency and scale of emergencies, the importance of which is constantly growing, is global climate change. Modern global climate warming causes an increase in the frequency of extreme meteorological and climatic phenomena and, accordingly, their environmental consequences [51]. An increase in the atmospheric humidification and groundwater level in turn also results in negative environmental changes, including an increase in the risk of hazardous floods, waterlogging, and coastal abrasion, and intensification of erosion processes, landslides, and mudflows. In certain extreme weather years, such processes in the basins under consideration become catastrophic and destructive. This is also manifested in accelerated soil erosion, death of crops, and surface water pollution by flushing products, destruction of engineering structures, and other negative environmental consequences that worsen living conditions and farming [52].

Years of extreme manifestations of erosion processes were distinguished by long-term data on the suspended sediment yield in 23 river basins. The criterion for the extremeness of the processes is the value of the modulus of the suspended sediment yield, with a supply not exceeding 15%. When considering the chronology of extreme events of the suspended sediment yield, their relationship with river flow extremals is established. The correlation coefficients between these parameters vary from 0.60 (Irkut: 0.62; Uda: 0.64; Chikoi—0.66) to 0.90 (Borzya), averaging 0.70 (Selenga: 0.70; Shilka: 0.75; Khilok: 0.77).

Based on the study of the difference–integral curve of water flow and suspended sediment in the basins over the last 50 years of the 20th century, the extremals of suspended sediment are associated with secular cycles of increased water content (Figure 3), which, taking into account the published

data, retrospectively extends the analyzed series until the beginning of the 21st century in [53]. In the Angara basin, phases of increased water content and, consequently, intensification of erosion processes were registered in 1906–1921, 1930–1952, 1959–1975, and 1983–1995 [54].

In the Selenga basin, an increased river water content and intense basin erosion were observed in 1959–1973 and 1984–1995. Although, the extreme development of erosion processes did not occur in the second period due to a sharp decrease in economic activity in the basins of several rivers in Buryatia, which was reflected in the graph of the difference–integral curve of suspended sediment of the Selenga (see Figure 11).

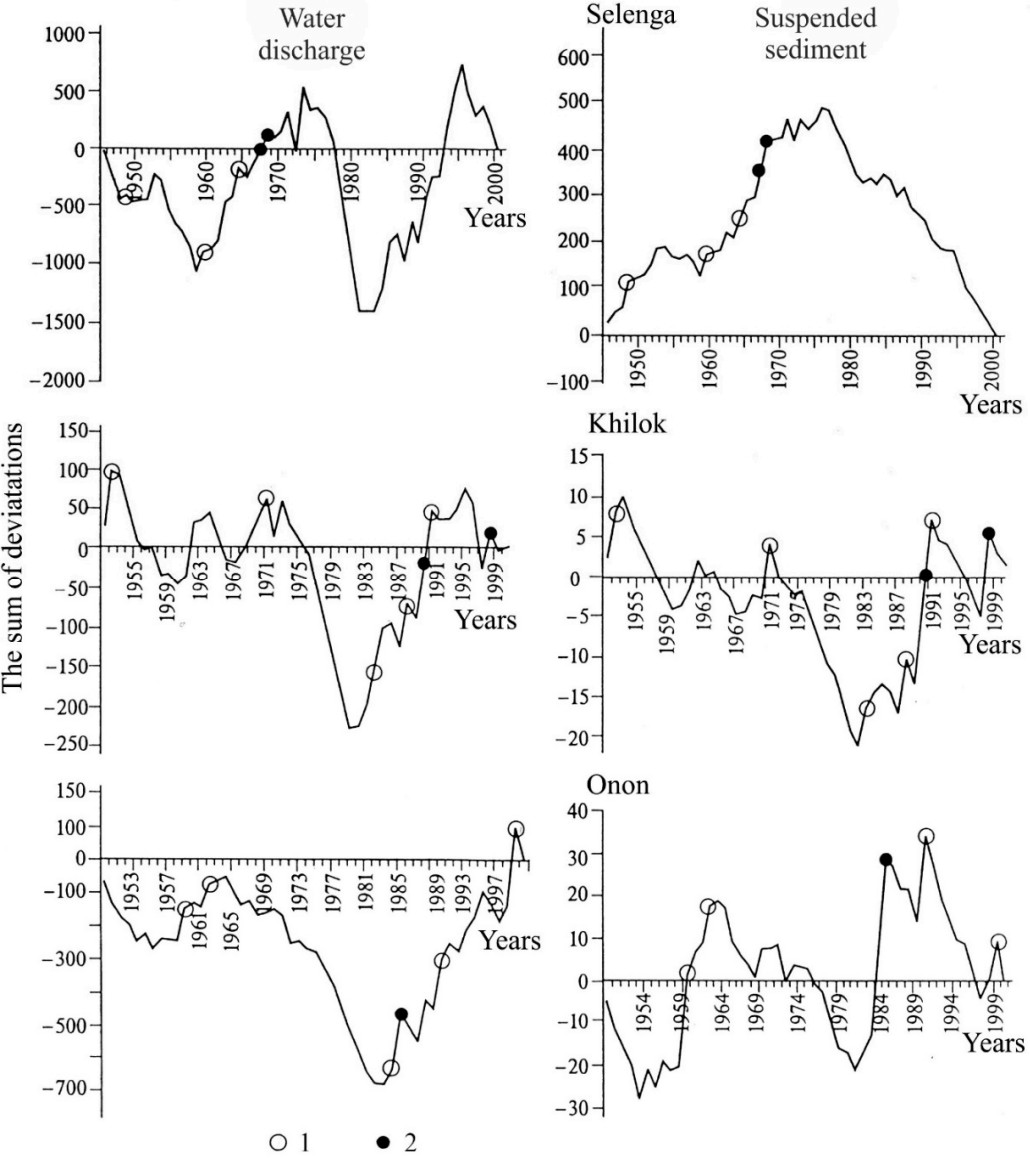

**Figure 11.** Difference–integral curves of water discharge and suspended sediment of the Transbaikalia rivers (1—extremals of suspended sediment yield, 2—the maxima).

In the upper reach of the Amur basin, high-water phases of the interdecadal cycles of river discharge fluctuations were observed in 1906–1910, 1932–1937, 1959–1964, and 1983–1991 [52]. Taking into account the recent years, the duration of the branch of increased water content can be extended to 1999 (see Figure 11).

The zero-phase progression of water runoff in neighboring river basins of the south of Siberia predetermines the formation of regionally significant extremals of the suspended sediment yield.

For Transbaikalia, a regionally significant activation of erosion processes, which covered the basins of the Barguzin, Uda, Chikoi, Khilok, Onon, Ingoda, and Shilka, was recorded in 1962. It was caused by positive anomalies of precipitation (Borzya, Ulan-Ude, etc.), and almost on all rivers coincided with high floods. The extremals of the suspended sediment yield confined to the last phase of increased water content of the 20th century and observed in 1985 (Uda and Onon), 1988, 1990, and 1998 (Khilok, Chikoi, Borzya, Shilka and Onon) are of regional importance for Transbaikalia.

To identify the hazard rate of erosion processes, we estimated the probability of suspended sediment yield formation of various intensities (Table 4), which allowed us to subdivide the natural risks of erosion process manifestation as: Acceptable (modulus of suspended sediment yield ≤ 15 t/km$^2$ per year), significant (15–30 t/km$^2$ per year), extreme (30–50 t/km$^2$ per year), and catastrophic (more than 50 t/km$^2$ per year) [26]. Of especially high probability of catastrophic erosion development is the Angara basin. In the Selenga basin, it is possible in small river basins formed by easily eroded soft grounds, as well as on the rivers of the Upper Amur basin with a probability of 4%.

**Table 4.** The probability of the erosion process development of various intensities in the south of Siberia, %.

| River (Locality) | Observation Period (Years) | Intensity of Erosion by Repeatability of Modulus of Suspended Sediment Yield, t/km$^2$ per year | | | | | |
|---|---|---|---|---|---|---|---|
| | | Very Weak, 0–5.0 | Weak, 5.1–10.0 | Moderate, 10.1–15.0 | High, 15.1–30.0 | Very High, 30.1–50.0 | Extremely High, >50 |
| The Angara basin | | | | | | | |
| Iya (Tulun) | 49 | 10 | 37 | 18 | 21 | 12 | 2 |
| Biryusa (Shitkino) | 51 | 33 | 45 | 20 | 2 | – | – |
| Uda (Alygdzer) | 33 | 21 | 43 | 21 | 12 | – | 3 |
| Ikei (Ikei) | 23 | 52 | 35 | 9 | 4 | – | – |
| Irkut (Smolenshchina) | 39 | 5 | 5 | 15 | 28 | 21 | 26 |
| The Selenga basin | | | | | | | |
| Khilok (Khailastui) | 50 | 76 | 18 | 6 | – | – | – |
| Chikoi (Gremyachka) | 48 | 35 | 29 | 19 | 17 | – | – |
| Selenga (Mostovoi) | 56 | 48 | 50 | – | – | – | – |
| Uda (Ulan-Ude) | 51 | 90 | 10 | – | – | – | – |
| Kuitunka (Tarbagatai) | 14 | 14 | 21 | – | 44 | 14 | 7 |
| The Amur basin | | | | | | | |
| Borzya (Borzya) | 34 | 82 | 12 | 3 | 3 | – | – |
| Onon (Bytev) | 52 | 44 | 38 | 12 | 6 | – | – |
| Shilka (Chasovaya) | 44 | 41 | 45 | 8 | 2 | – | 4 |
| Ingoda (Atamanovka) | 23 | 32 | 42 | 16 | 10 | – | – |

In accordance with the background forecast for fluctuations in the flow of Siberian rivers, it is possible to outline the beginning of the next phases of increasing river water content and the intensity of erosion processes [34,52]. Attention should be paid to the possibility of using the consistency of catastrophic storm floods on Siberian rivers with solar activity cycles in the forecasts [53]. A close relationship between erosion processes (soil wash on croplands) and solar activity has been established on the Russian and West Siberian plains with a correlation coefficient of 0.76–0.97 [54]. We should note that in 80% of cases, the maxima of suspended sediment yield on the rivers of southern Siberia gravitate to the minima of the solar activity cycles [34].

One of the main climatic prerequisites for the implementation of an extreme fluvial event is the storm regime of precipitation. Besides flooding, heavy rains caused a dramatic intensification of the basin soil, gully, and channel erosion. In the case of large daily precipitation of high intensity, mudflows can also form in small river basins and temporary streams. The recurrence of such phenomena in the Angara basin is 7–10 years [34]. They occur in summer, when 75–80% of the annual precipitations fall. The recorded maximum precipitation for one month exceeds 150 mm. The largest monthly precipitation was recorded in Irkutsk (303 mm in June of 1938). The maximum daily precipitation ranges from 59 (vil. Rasputino) to 84 mm (st. Polovina). One shower lasting only a few hours sometimes brings more than 40 mm of precipitation. The shower intensity in this case is 1–2 mm/min. The maximum rain intensity on 3 August 1959 in Bayandai reached 4.1 mm/min. The precipitation depth in this case exceeds 50 mm, and sometimes reaches 70–100 mm. So, in Irkutsk on 6–8 July 2001, the amount of precipitation was 98.1 mm, and in 1939, a record amount of precipitation (119 mm) fell here during the rain on 17–19 June. The year 1906 is distinguished by the abundance of rainfall [55]. In the 19th century, 1869 was characterized by an unusually heavy storm in the Baikal region. Its consequences were so serious that they caused a special expedition of a member of the East Siberian Department of the Russian Geographical Society A.N. Orlov for the study of floods as consequences of especially large showers in the Baikal region [56]. In the 20th century, two periods with a very high hazard of showers are clearly distinguished in the Upper Angara region. This is, first of all, the period from 1936 to 1962 and period from 1991 to 2002, during which active growth of the gully network was recorded. Thus, an increase in the length of the bottom of one of the gullies in the Zakuleisk erosion massif was 19 m [31].

As a scenario of a catastrophic fluvial event, we considered the consequences of showers in June of 1960 [40,49]. In the Bokhan district as a result of a storm on 18 June, 1310 ha of crops were damaged, and on 23 June 1960, 1819 ha. In the Osa district, as a result of a hail shower, part of the crops was covered with soil, and in some areas the plants were washed with water flows. In total, 4426.3 ha of crops were seriously damaged and partially lost [40]. Storm rainfalls passed through areas that did not exceed 500–700 km$^2$. They were the cause of mudflow formation on small streams and blind creeks [49].

According to experimental observations for one storm rainfall, up to 3 kg of fine earth is removed from a burning area of 1 m$^2$ [31]. In the area of the Bratsk reservoir, erosion grooves with a depth of 7–15 mm occur on the burning sites, and traces of fresh sod disruption were noted. In some cases, gravel is also involved in the movement. The aspect has an influence as west- and east-facing slopes are washed more intensively [31]. According to the calculations on the Irkutsk-Cheremkhovo plain, soil losses from storm wash on agricultural lands are 8–20 t/ha per year and the maximum values of basin erosion vary from 60 to 130 t/km$^2$ per year [57].

In the Selenga basin, periods of high moistening favorable for the development of water soil erosion were noted at the end of the 17th and the beginning of the 18th centuries, as well as in 1780–1790, 1805–1830, and 1862–1875. [58]. Unique information on the catastrophic manifestations of soil erosion processes caused by extreme high-intensity storm rainfalls is contained in the works by N.A. Kryukov [59]. So, on 11 August 1886, about 75 mm of precipitation fell in the Petrovskii Zavod in two and a half hours; on 14 July of the same year, 53 mm of precipitation fell in Chindant. Like this, on 13 August 1888, similar rain in the village Mukhortalinskii washed away wheat crops from 13 ha, carried away the fertile layer, and formed deep gullies on croplands. Streams rushed so swiftly along the narrow steep streets of the village that they demolished the hedges. Even greater was the destruction of 20 July 1889 on the steppe slope in the valley of the Chikoi river, when the water carried yurts, barns, and cattle. In the basin of the Podbaikhora river (a tributary of the Chikoi) in 1886 and 1887, the humus soil horizon over an area of about 165 ha was washed away by showers from western slopes. The stony material taken from the mountains covered up large areas of hay meadows; at the same time, the water produced many deep potholes stretching almost to the summits of the ranges.

N.A. Kryukov [59] reported on the extreme condition of the sloping croplands on the right bank of the Dzhida, subject to intense sheet erosion: "New arable lands turn black from afar, and old ones turn white or yellow, since water and winds carried away the upper fertile layer and they are no

longer sown. Summer rains wash away the mass of land from these slopes and take it to the Jida riverbed. The sediments of this river have especially increased in recent years, as a result of which the river often began to change its channel and in the eyes of one generation it was divided into several channels and branches" [59]. He also noted the deplorable state of croplands in the Khilok valley, especially in the vicinity of the village Bichura, where there are many potholes and gullies, which, according to old-timers, appeared during a period of high humidity around 1862–1875. Intense channel deformations with a catastrophic destruction of the coastal slopes of the Selenga, Chikoi, and their tributaries were characteristic of this high-water period. They were especially active in 1862, 1869, and 1875. Many riverine settlements suffered from them [59].

Thus, during an extreme fluvial event, we can observe supercritical removal of a large volume of soft ground. At the same time, the rate of erosion processes can increase dramatically, and cause soil erosion and degradation of agricultural land; the size of erosive landforms increases, and new cuts occur, crops are damaged, and roads and buildings are destroyed.

### 3.6.1. Debris Floods

Many river basins in the south of Eastern Siberia are characterized by an increased debris flood hazard. In the 20th century, powerful water streams were formed during storm rainfalls in local areas in the Angara basin in 1958, 1960, and 1962; in the Selenga in 1988; and in the Upper Amur in 1988, 1997, and 1998. In the Selenginskii highlands, debris floods occur during short-term intense rains in July–August. Debris and mud-stone floods are characteristic of many small rivers. They were observed on the Kuitunka river near the village of Tarbagatai in 1914, 1950, and 1960. Significant debris floods, causing significant material damage, took place on 13 August 1965 on the Borgoi river and 11 June 1968 on the Gryaznukha river near the town of Kyakhta.

Debris flows are a characteristic regional feature of exogenous relief formation in the steppes of Central Asia, both at present and in the Holocene [16,60]. Here, on a semi-mountainous terrain, water flows arising during storms on the slopes have a high transporting capacity. On steep slopes with a sparse grass stand, fine earth is immediately carried down. Water storm flows are especially active in the narrow steeply falling bottoms of the pads, where their width can reach 7–12 m and a depth of 30–50 cm. We observed such flows in the Krementui pad, the position of which is shown in Figure 1. The high levels of the lakes correspond to the phases of fluvial activity in the Holocene. In the middle part of the bottom of the Krementui pad, in the section in the range of 0.95–2.1 m, fluvial sediments are distinguished, which, in our opinion, were formed in a very short period of time as a result of an "explosive ejection of material" from the upper reaches of the valley. Against this background, two periods of their particularly high activity are registered: In transition from the Atlantic to the subboreal period and in the 14th century [60].

The probability of the same hydrometeorological phenomenon in the same catchment basin for a short period of time is very small, since the frequency of occurrence of intense precipitation often does not coincide in time and space. This situation developed in 1962, when debris flows in the mountainous surroundings of Lake Baikal formed in different places from June to August.

Debris flow activity in the Baikal rift zone has been researched over the past three centuries [61]. In this period, 1871 and 1971 are distinguished by the area covered by debris flows and the clear preservation of their traces, indicating a significant power of flows. Trees damaged in 1871 were found almost throughout the Stanovoi Highland: From the Baikal range to the northeastern end of the Kodar Range. In 1971, various streams, from powerful mud-stone to weak water-stone, were formed in the territory from the East Sayan to the North Muya Range during prolonged anomalously stormy rains in July. According to the predominantly dendrochronological analysis, the debris flow areas in the Baikal region can be most clearly restored and often appear in three areas: In the East Sayan (Tunkinskie Gol'tsy), in the mountainous surroundings of the southern end of Lake Baikal, and in the Northern Cisbaikalia, including the Stanovoi Highland. Debris flows are very widely developed in Baikal mountains. The greatest destruction of engineering structures occurred in the 20th century in

the Southern Cisbaikalia, on the coast of Lake Baikal, and along the Khamar-Daban ridge, which has been developed by man.

Mud-stone and water-stone flows were observed here. Significant floods were registered in 1915, 1927, 1932, 1934, 1938, 1959, 1960, 1962, and 1965. The most powerful of them, which caused great material damage, was a debris flow on the Slyudyanka river in 20 June 1960 [62]. In other mountainous areas, traces of rare, isolated, and relatively thin debris flows are found.

There is a periodicity in increasing and decreasing areas of debris flow manifestation. In the period of 1910–1931, a decrease in debris flow activity is noted (only few cases of debris flows).

Since the mid-1930s, there has been an increase in the positive trend of atmospheric precipitation fluctuations, while solar activity is increasing. In general, debris flow activity in the Northern Baikal region in 84% of cases falls on the branch of a decline in solar activity and in 16% on the branch of its rise.

It is established that a significant part of the debris flow manifestation in the same years was noted in two or three areas of their usual manifestation (for example, debris flows of 1935, 1942, 1952, 1960, 1962, and 1971), which is associated with the large hydrometeorological events covering large areas in Cisbaikalia. Most of the debris flows were formed due to the creep of ground [63]. On the slopes of the river and stream valleys between the cities of Baikalsk and Slyudyanka in 1971, about 150 creeps of ground with an average volume of 3000 to 10,000 m$^3$ were formed. The total volume was at least 1 million m$^3$ of unconsolidated material (based on aerial photographs of 1972). After the catastrophic debris flows of 1971, their activity sharply decreased.

In the last 40 years, debris flow activity in the Baikal rift zone has practically not been manifested, with rare exceptions. However, it was established that long periods of a relative calm of debris flow activity were always replaced by periods of their dramatic intensification. Additionally, since 2014, debris flow activity has increased again. On the night of 27–28 June 2014 in the village of Arshan (Tunkinskii district, Republic of Buryatia), storm rainfalls gave rise to debris flows of two types: A water-rock flow on the Kyngarga river, heading from the southern slopes of the Tunkinskie Gol'tsy Range (East Sayan), and mudstone flows along the valleys originating in the cirques of the southern slope of the range [64] (Figure 12).

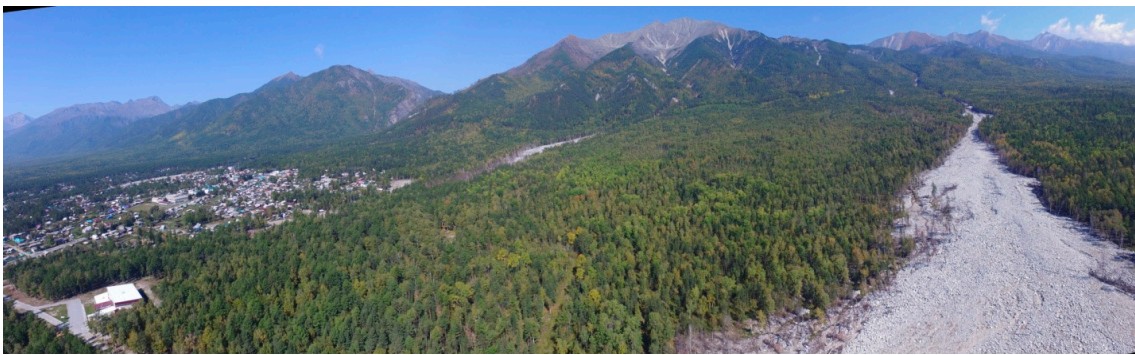

**Figure 12.** On the right is the frontal part of the mud-stone flow that passed along the river Vtoraya Shikhtolaika, on the left is the vil. Arshan (51°54′29.2887″ N, 102°26′50.9295″ E). The image was taken on 9 September 2016.

The volume of sediments washed from the cirques of the southern slope of the Tunkinskie Gol'tsy Range amounted to about 2 million m$^3$. An extensive cone was formed on the plain. In the lower parts of the Kyngarga river fan, with an abrupt decrease in the flow velocity of the water saturated with suspended load, they accumulated on top of the soil cover with a thickness of up to 2 m (Figure 13).

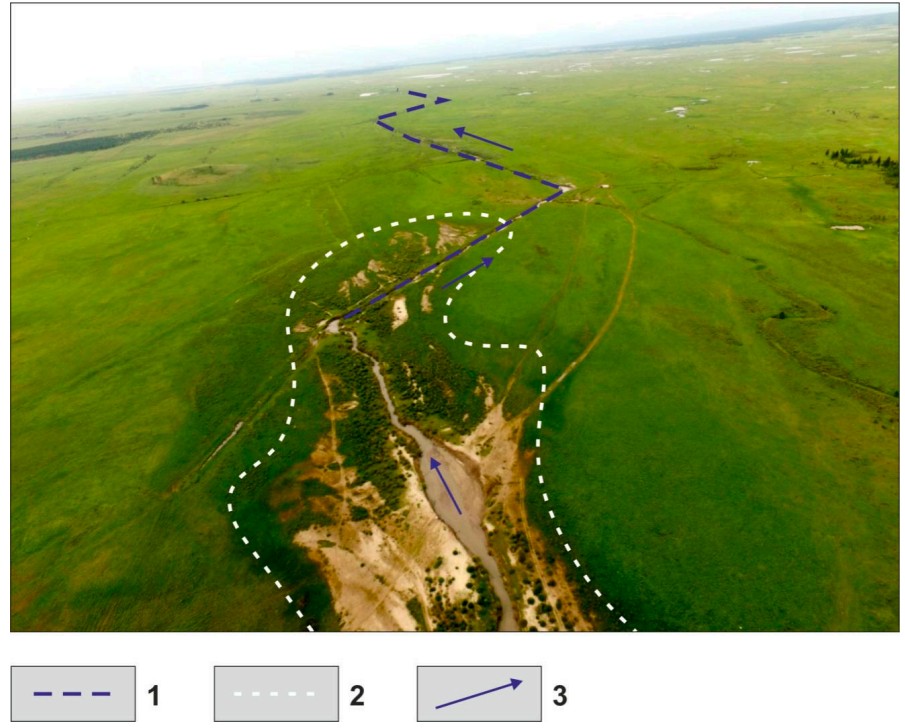

**Figure 13.** Zone of accumulation of suspended load on the Kyngarga river where it enters the dike. Images were taken on 12 July 2017, from the Phantom 3 quadcopter, at 234 m (51°51′5.36″ N, 102°27′20.09″ E). 1: boundary of accumulation of suspended load; 2: dike; 3: direction of the river.

One year after the passage of disastrous debris flows in the area of the village of Arshan, on 14 July 2015, the Kyngarga river developed a water-rock flow again (Figure 14).

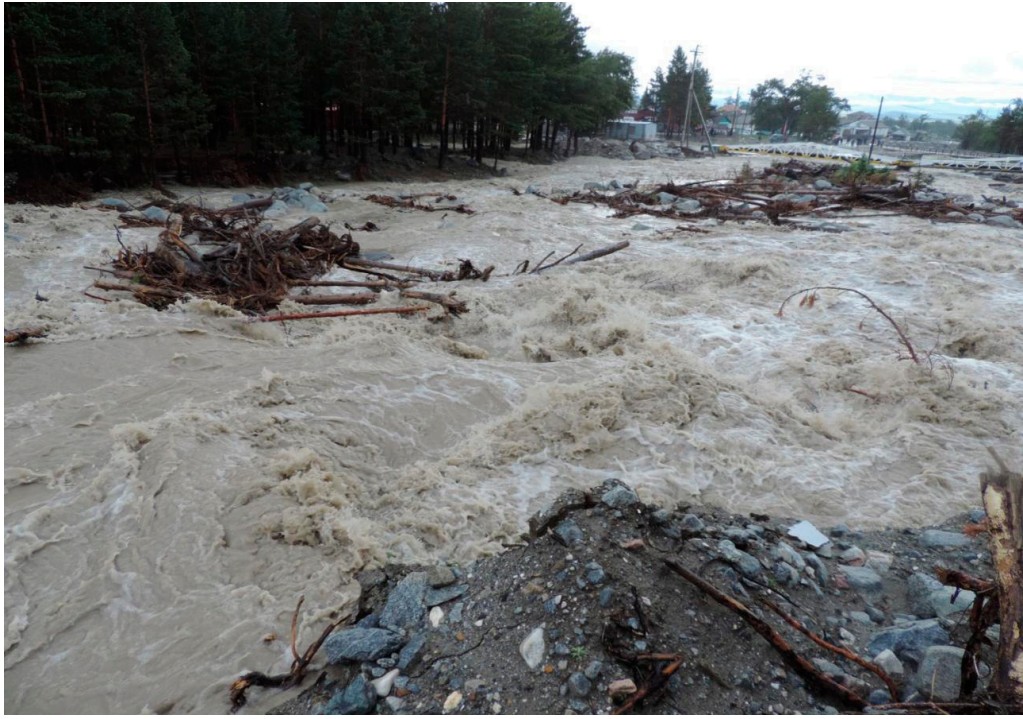

**Figure 14.** Water-rock flow on the Kyngarga river at the 14 July 2015 peak maximum, time: 11:45.

This event occurred after a long drought in the south of Eastern Siberia. During the passage of water-rock flow on 27–28 June 2014 on the Kharimta and Mal Kharimta rivers, there was a restructuring of the river network. After the mudstone mass clogged the channels of these rivers, the water flow shifted to the left of the Mal Kharimta and moved along the previously dry nameless channel, toward the village of Khurai-Khobok, where it inundated the depressions along the motor roadside. On its way, it crashed into boulder deposits (Figure 15).

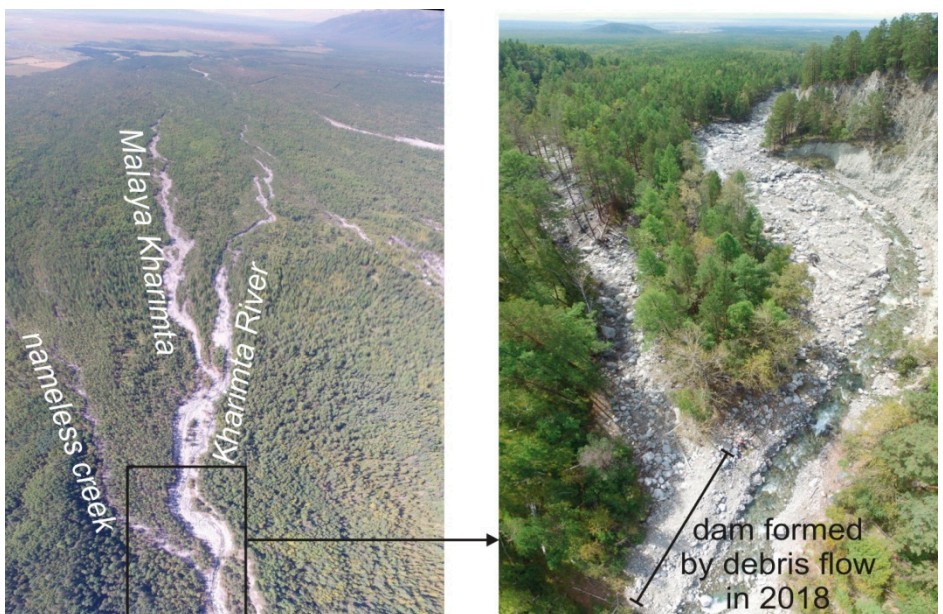

**Figure 15.** Reconstruction of the river network after the descent of debris flows on the Kharimta and Malaya Kharimta rivers in 2014 and 2018 (Tunkinskie Gol'tsy Range). 51°54′1.44″ N, 102°30′55.04″ E. The image was taken on 1 September 2018.

The recent pre-mudflow situation on the rivers near the city of Baikalsk occurred on 28 July 2019, when a cyclone caused powerful floods on the northern slope of the Khamar-Daban ridge [64]. A technological reinforced concrete bridge on the Solzan river was destructed (Figure 16C). The communication lines that were located on both sides of the bridge were preserved.

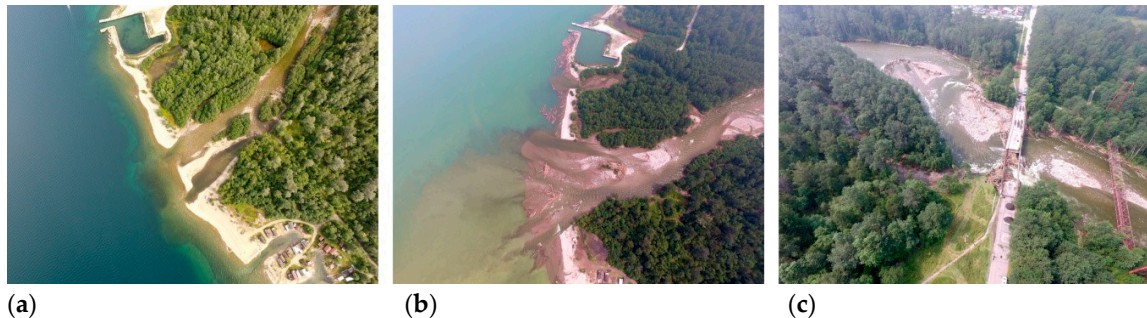

(**a**)  (**b**)  (**c**)

**Figure 16.** The consequences of the flood on the Solzan river. (**a**), (**b**) channel morphodynamics at the mouth of the river. Coordinates: 51°31′46.6418″ N, 104°9′27.5627″ E. Images was taken: (**a**) 23 August 2017, (**b**) 31 July 2019; (**c**) the destroyed technological bridge. Coordinates: 51°31′24.6506″ N, 104°9′43.3299″ E. Images was taken on 31 July 2019.

Above it, a bend of a divided meandering channel was formed. The stream here bends around a large stable side wall, located near the convex left bank and thus reflecting from the opposite bank, passes right at the location of the intermediate bridge support. A powerful water stream eroded the base under the intermediate support, thereby causing its deformation and, as a result,

the destruction of the right-bank part of the bridge. During the flood, a stream from a channel located near the right bank probably developed further, increasing the capacity of the main channel stream. In 2017–2018, the channels of many rivers were cleared of woody vegetation to avoid tree-shrub congestions. This timely work played a positive role, and channel flows of water were carried into Lake Baikal without delay. The valleys of these rivers began to overgrow with woody vegetation after 1971. This was a year of numerous debris flows and floods on the territory of Cisbaikalia. The river valleys have been completely cleared. Over the past 48 years, they have mostly been overgrown with poplar, aspen, and willow and alder. A thin soil horizon formed. During the flood in the river valleys, the soil was blown into Lake Baikal and boulder and pebble deposits appeared on the surface again. On many rivers, channel incisions were recorded in the bottoms of valleys to a depth of 1 m.

Significant changes have taken place in the estuaries. When comparing surveys performed in 2017 and 2019 on the Solzan river, we found that: (1) A large number of transported and suspended sediments were brought into Lake Baikal, the latter did not accumulate in the coastal shelf zone after four days (see Figure 16A,B); (2) the islands at the mouths of the rivers completely cleared of vegetation; (3) creeks eroded islands and beaches formed by pebble-gravel-sand deposits; and (4) woody vegetation was carried in Lake Baikal, which was washed ashore. At the mouths of other rivers, a similar situation formed.

The catastrophic debris flows of 2014, which had no power in the historical past [65] as well as subsequent extreme events, including floods that occurred in the southern basins of the area under study in 2019 and caused floods in some places [66], mark the beginning of a new synoptic situation. In the near future, we should expect a sharp activation of exogenous geological processes associated with active cyclonic activity in the south of Eastern Siberia.

### 3.6.2. Activation of Exogenous Relief-forming Processes of the Iya River Valley

In the summer of 2019, catastrophic floods took place on a number of rivers located in the left-bank part of the Angara river basin as a result of intensified cyclonic activity. They caused human casualties and significant damage to the national economy. In total, 1311 houses were taken away by flood waters, 4191 houses were declared unsuitable for living, and 3846 had to be repaired. The preliminary damage amounted to 35.152 billion rubles; 10.8 billion rubles of this was loss from lost housing, and 20 billion rubles from damage to social, administrative, and infrastructure facilities [67]. The cities of Tulun (Figure 17) and Nizhneudinsk suffered mostly from the floods, located in the Presayan piedmont depression and Iiskii and Udinskii rifts. The sources of flooding were, respectively, the Iya and Uda rivers, originating on the northern slopes of the East Sayan.

Inspired by these events, we carried out field research in a number of sections of the Iya river valley in various geodynamic settings, including during the flood (August–September 2019). We analyzed the geomorphological conditions of the territories as one of the factors of flood formation and flood events. The intensity, types of manifestation of exogenous geomorphological processes activated by floods, were estimated.

The Iya river basin within the area under research crosses a number of heterogeneous geomorphological surfaces— the mountain-folded region of the East Sayan uplift to the trap regions of the Central Siberian plateau. Geological and geomorphological conditions provided a variety of morphodynamic channel and floodplain types within the river valleys [68].

The Iya river in the mountainous region has predominantly adapted and incised channel types; within the flat-platform part of the basin, when crossing sandstone fields, a broad-flood meandering channel with free and adapted bends is developing. When the river crosses the traps of the Triassic age, incised and adapted channel types have formed.

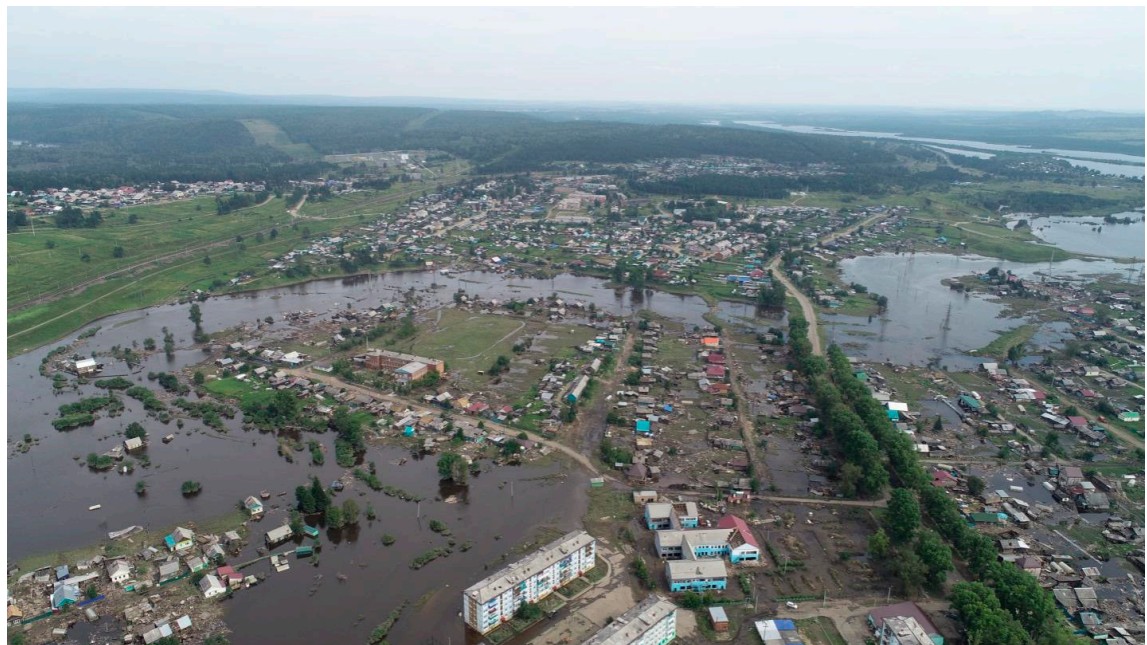

**Figure 17.** Panorama of a district, Tulun. Coordinates: 54°32′38.9315″ N, 100°37′41.9411″ E. The image was taken on 2 August 2019.

Within the broad floodplain area of the Iya river, an additional factor for flood subsidence and distribution in this area was the variety of floodplain types, in particular, the valley-island type and the general predominance of plane accumulative areas (Figure 18). Water-flow retardation was also facilitated by the slight predominance of zones of erosion and wetland-accumulator areas within the floodplain.

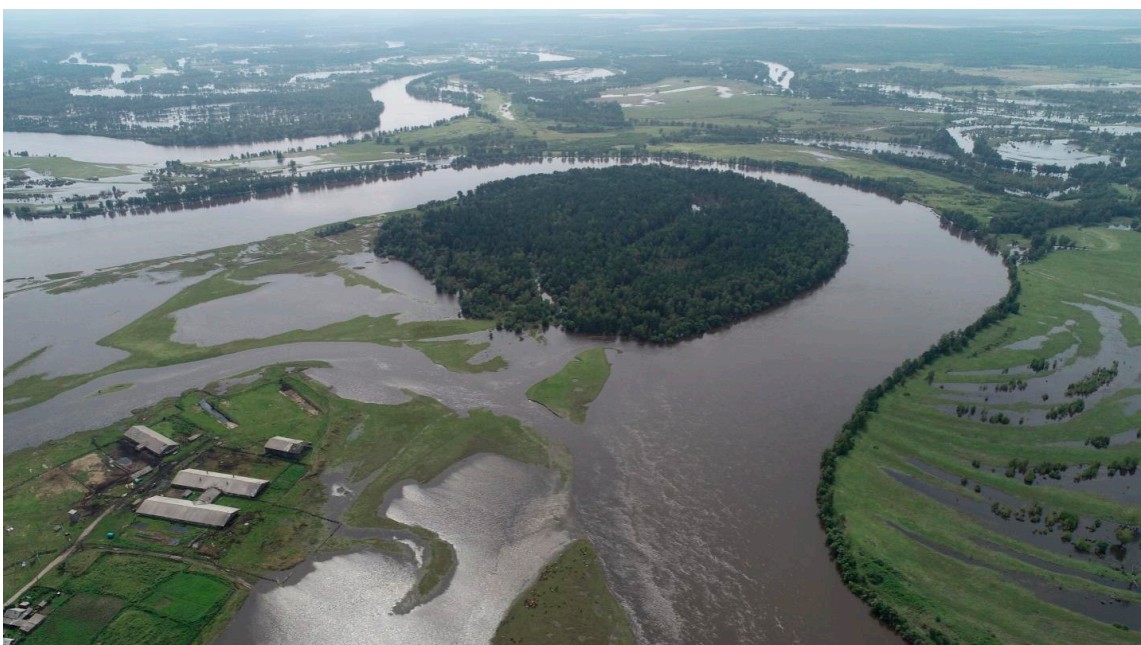

**Figure 18.** The flooded negative forms of the floodplain topography of the broad floodplain section of the Iya river (the village of Gadalei). Coordinates: 54°24′27.4291″ N, 100°43′23.1787″ E. The image was taken on 2 August 2019 (second wave of the flood in 2019).

Within the downstream section the Nizhnii Manut, Tulun, the nature of the flooding was affected by the alternation of different types of channel and floodplains, and change in the floodplain capacity, which affect both the types of interaction between the channel and floodplain flows and the types of erosion-accumulation processes. Thus, dacha villages, located in the spurs of adapted bends, upstream of the Gidroliznyi district, were in a kind of trap. Here, the narrowing of the Iya river valley, after the broad floodplain of the Gadalei expansion, served as an additional factor that strengthened the stream, which literally dared all the buildings. Within the city of Tulun, the Iya river forms a macro-bend, the spur of which is a floodplain. Downstream, the river has adapted and incised channel types [69].

The nature of the distribution of flood waters within the adapted section of the channel was affected by (a) its location between two broad floodplain river stretches, and (b) the geometry of forced bends within a narrow valley. The combination of these factors provoked the flooding of a vast territory with disastrous consequences for the population of the Tulun district.

During the flood (August–September 2019), the activation of slope processes was recorded, which were most extensively and extensively manifested in the mid-mountain and foothill areas of the East Sayan uplift [70]. The steep slopes are indicative in this regard, where various slope processes have developed. In the upper parts of steep slopes, mudslide formation was recorded; in the middle parts they were joined by the processes of caving. In the lower parts of the slopes interacting with the river flow, a sufficient amount of slope material initiated the formation of a skeletal floodplain.

In the lower parts of the slopes on benches of floodplains and low terraces, mainly the processes of creeping of the sod layer developed, which often serves as a kind of visor and prevents the destruction of the benches by the river flow.

Within the flat part of the Iya river basin and proximately in its valley, the intensity of the process manifestation is much lower; however, due to the population density and location of many objects here, the development of such processes can be dangerous for humans and their economic activities.

In the immediate vicinity of the city of Tulun, the activation of erosion processes on coastal benches was recorded, in particular, the formation of subsidence fracture, caving, sliding of blocks, and linear erosion processes. In the coastal zone, we registered small erosion forms up to 40 m long, 2 to 20 m wide, and 2 m deep.

Thus, the rainfall and the subsequent floods caused the activation of exogenous geomorphological processes, including those dangerous to humans. For the mountainous area, these are landslides, creeps, and mudflows. For the plains, these are waterlogging, suffusion-subsidence phenomena, caving, and creeping of soil on coastal benches, as well as the possible activation of cryogenic processes. The processes can be categorized into two groups according the nature of intensity: (1) Processes of instant response (which have already occurred, as a rule, these are processes of short-term manifestation), and (2) processes with delayed action, the active manifestation of which can stretch for several years (but the conditions for their formation have already been created).

## 4. Conclusions

The research showed that the functioning of erosion-channel systems in the south of Eastern Siberia resulted in an intensive redistribution of sediment in river basins. All groups of water flows, covering slope, gully, and channel elements of basin systems, actively participate in sediment movement. The climatic regime of surface waters, uneven in time, determines the pulsating nature of the matter transport and formation of extreme fluvial events that significantly complicate the ecological situation in the basins under consideration.

During these events, explosive ejection of a large volume of unconsolidated sediments occurs. At the same time, the velocity of erosion processes can increase and cause soil erosion and degradation of agricultural land. The size of erosive landforms increases dramatically, thus forming new incisions, damaging crops, and destroying roads and buildings. Of particular danger to human life are floods and debris flow floods.

Since the 1990s of the past century in the south of Siberia, a downward trend in the suspended sediment yield has been noted, which allows us to talk about an improvement in the ecological situation in the basins under consideration, due to a decrease in the intensity of erosion processes against the background of conservation of agricultural land. In this case, it is necessary to emphasize a reduction in the flow of pollutants into Lake Baikal as an environmentally important consequence of the changes. The exception is zones with high rates of storm wash in areas of felling and fires.

At the same time, in 2014, a new cycle of an increased frequency of extreme fluvial events on the East Sayan and Khamar-Daban rivers began. The catastrophic consequences of these floods and debris flows necessitate the adoption of urgent measures to protect against such natural disasters. Among these measures, first of all, it is necessary to exclude building development of dangerous territories and improve the accuracy of forecasts. To overcome the environmentally adverse effects of soil erosion, it is necessary to carry out special anti-erosion measures, using international experience [71,72].

**Author Contributions:** O.I.B., E.M.T., M.Y.O., S.A.M. conceived the idea and developed the structure of the paper. A.V.B. and S.A.T. performed calculations and mapping of potential erosion. All authors took part in the research of fluvial processes and the preparation of the paper. All authors have read and agreed to the published version of the manuscript.

**Funding:** This work was done within the framework of the Integration program "Fundamental research and breakthrough technologies as a basis for advanced development of the Baikal Region and its interregional relations" (No. 0341-2017-0001) and with a financial support from the Russian Foundation for Basic Research (17-45-388070-p_a).

**Acknowledgments:** We thank Inna Zlydneva for assistance with translation the manuscript and for comments that greatly improved the manuscript.

**Conflicts of Interest:** The authors declare no conflict of interest.

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
