# Peer review of "The Functioning of Erosion-channel Systems of the River Basins of the South of Eastern Siberia"

_geosciences, doi:10.3390/geosciences10050176_

Round 1

Reviewer 1 Report

The paper covers the highlights and graphical abstract as well as it reports interesting data (even if poorly analyzed). The methods of investigation are well structured and the discussions are organized consecuely very well. In order to improve the work for publication, data analysis could be improvedand  in the introduction clarify some aspects as well.

The manuscript deals with a relevant topic, which should be of interest to the journal's readership. It presents an interesting data set concerning geomorphological processes. However little changes are necessary before the publication.

In the introduction section, I don't quite see the explanation about the "significant contribution" of this work. Here the authors can work more.

The paper should be improved with information about rainfall trends, land use changes, local channel slope changes and longitudinal channel profiles.

Other statistical indexes could be necessary to explain the relation between sediment, channel shape and vegetation cover.

An rearrangement of the maps (figure 1 and 2) is necessary.

Discussion section is too long, some sentences can be avoid

The article, in the bottom, reports an interesting investigation about interaction between channel morphology and land-use changes; you can assess it and report in the references

Adjustments in channel morphology due to land-use changes and check-dam installation in mountain torrents of Calabria (Southern Italy). Earth Surface Processes and Landforms. 42 – 14 pages 2469-2483 by FORTUGNO D., et. all. (2014):

Author Response

Dear Reviewer!

      The authors are grateful to you for viewing our manuscript and valuable comments aimed at improving the article for publication.

To increase the significance of the work in the introduction, the rationale for this study was added:

In the Conclusion, attention is paid to the practice of protective measures, an insert and a link to the Italian article recommended, are given

In relation to mountain rivers, it is also important to use the experience of creating anti-erosion structures used in other mountainous regions of the world (Italy, China, etc.).

Figures 1 and 2 improved.

Line 61 agricultural mastery inserted. The period of intensive agricultural development in the south of Eastern Siberia is about 130 years. It is associated with the construction of the Trans-Siberian Railway, then in the Irkutsk province agricultural area increased by 3.5 times. Plowing slope lands caused widespread development of soil erosion. With the catastrophic manifestation of these processes, the peasants were forced to transfer the arable land to a fallow for many years. In the Baikal region in the 19thcentury up to 15% of arable land was abandoned due to washing out.

A new powerful impetus to the development of erosion processes was given in the postwar years during the period of mass development of virgin and fallow lands (since 1954). Currently, in the south of Siberia, the natural soil and vegetation cover has been changed to 70–90% of the territory. As a result of plowing the slopes, pasture digression, it significantly lost its protective anti-erosion properties, as a result of which accelerated erosion was widespread when the washing out is not compensated by the soil formation process.

Currently, due to the forced conservation of agricultural land, noted since the 90s of the last century, soil loss from erosion has significantly decreased (Bazhenova, Kobylkin, 2013).

Reviewer 2 Report

Detailed comments to Bazhenova et al. (766753)

Comment by section, and line #.

Abstract

The abstract is concise and provides a good summary of the work.

Introduction:

The introduction is clear and presents adequate information about the objective as well as the reason for the selected area.

Material and methods

86: The authors state that computational models ar used.

102: The RUSLE does not contain just four factors, it consists of six. Please do not summarize factors as these is misleading e.g. topography = slope length and slope steepness, land cover = vegetation cover of the landscape, land use = erosion control practice… These factors do not describe erosion, rathe tan soil loss, in this primarily by water. It would be wise to explain why the RUSLE has been chosen for the study, and not e.g. MUSLE or others?

Additionally, please state the source of these input parameters. Where did you receive this information? How reliable are the sources and what role play uncertainties? You have only presented the source for topographic information in this chapter.

Results and Discussion

125: what is your threshold for being “significant". Also line 124-127 repeats the same content…

163: How do you conclude climatic fluctuations? – Monsoon rains (153) are yearly event, where do you see fluctuations of climate?

181, 184: please use uniform units for erosion. In line 142, 143 you have used t/km2, now you use t/ha. For both you can yr1 instead “per year” e.g. [t ha-1yr-1]

186-187 again use same units. Now you use m3/ha. Why not convert it to tons with the according soil density?

189 -and then mm/year. I urge to use one unite as possible (you can still provide another format in bracks). For readers it is a faster way of comparing soil erosion rates if a consistent unit is used.

206 It appears that all the results are significant. Please state which ranges are really significant (and why) which are intermediate, and which are neglectable.

274, 280, 290 etc. : “ a large number of erosion form – per unit are… what unit are you refereeing too?

307: it is not clear to me if you have measured these intensities or if you are refereeing to literature source (which is then not cited here).

Conclusion:

Provide a suitable summary of key aspects, but of course cannot provide a fuller detail due to the comprehensive content.

Figures and Figure Captions:

In general, please use meaningful and informative text to the figures.

Figure 1: Please indicate source of the map. Clean the map of all uncessary information which will not be used in the manuscript text. Use a larger font for lat and long and the scale bar.

Figure 2: Instead of describing the colour code in the caption, write the description in the actual legend (A and B). The title of the y axis should be clear on the side of the graph e.g. [t km-2yr-1]. Ideally the numbers and descriptions of the graph would match a map (fig. 1), for easy allocation of the varies columns, or do write the names instead of numbers

Figure 3: The erosion zoning of the map is good. Yet , the numbers 1-6 of the legend do not tell any useful information. Please provide more information what they mean. Do the represent a certain range of erosion? Line 218-226 appear to be the missing description for Figure 3. It this is the case, please use the values e.g. 1: <1 t ha-1 etc. in the legend.

Figure 4: The coloured bar legend is missing the unit. Please add, in order to provide adequate information (ideally the same unit for the whole manuscript). Reading the black coloured names on a dark background of the model is difficult – change to white , or other contrast rich colouring.

Figure 5: There is no scale  and no clear indication of the location of the picture. Could be references to the Fig. 1?. A does not provide a direction nor the position where the picture was taken. Please provide cleaner and higher quality Figures. The additional description (e.g. a, b, c) would be overall better if you frame it and fill it with a pattern and include a legend, instead of writing three times a etc. And again black letters on a dark background are hard to read, for most readers.

Figure 6. There is no citation in the figure captions in regard of image source. If you use a google earth image, that has not been “cleaned” of all the unnecessary information (google earth logo, scaling tool etc) than at least use a Latin letters. I would suggest to adequately clean the picture of all unnecessary things, provide more contrast rich writing (white intead of black) and the state the additional information (correct position, source) in the figure captions.

Figure 7: Please see my comments above. This is unexpectable. There is no scale, the picture has been stretched, no position, a city name in Cyrillic , no complete citation in the figure captions. So please revise this, and your other figures in order to fulfil at least the standard quality necessary in order be published in a peer-reviewed journal.

Figure 8: This is a wonderful, graphic. A slight improval, would be to use change the legend from numbers to words e.g. 1 would be “Floodplain”. Second use the title (for a, and b) above the two graphics. Third, explain the difference of state I and II between the two pictures in the figure captions. You can also indicate the time difference you present in I and II.

Figure 9: The thre time sections are not explaiens – use an adequate legend (1). Stack the graphics a and be like Fig 8, than the graphic would be large in the manuscript and better readable to the audience.

Figure 10: I am not sure if red is the useful colour for this rather dark graphic. Why is this satellite information not combined with Fig. 9? What differences I am supposed to see in the individual rectangles?

Figure 11: Please re-order the the indifcual graphes and number each and every one of them (a, b, c etc…) and provide a adequate description for each. Add a legend for the two dot types.

Figure 12: Okay.

Figure 13: Okay, please change legend form numbers to words.

Figure 14.: Okay, Position?

Figure 15: Okay

Figure 16: okay

Figure 17: okay (position?)

Figure 18: Okay (Position?

Figure 19: Position and date missing.

Figure 20: Position and date missing.

It might be wise to combine Fig. 20, 19, 18, 17 to one graphic to save some space in the journal. The importance of these oversized individual pictures are in question.

Author Response

Dear Reviewer!

      The authors are grateful to you for viewing our manuscript and valuable comments aimed at improving the article for publication.

Line 86. The authors state that computational models are used.

The authors of the article have been studying the slope runoff for many years using a wide range of methods for assessing the speed of processes. The basis of the research is the data of field experimental observations of soil erosion and erosion at hospitals of the Institute of Geography. B. B. Sochi SB RAS, in which the authors took a direct active part [Bazhenova et al., 1997; Bazhenova, 2018]. The observation period at the sites is 15-18 years.

In addition to field measurements, to determine the potential flush, we used a technique developed in the research laboratory of soil erosion and channel processes at Moscow State University [Larionov, 1984]. The possibility of using it to establish the average annual flushing rates in the southern regions of Eastern Siberia is confirmed by field measurements of the rate of deluvial processes ([ Bazhenova, 1993]. The correlation coefficient of the measured and calculated velocities is quite high (0.86 + _0.11). The method is based on the universal equation of soil loss as a result of vnevogo flushing [Mitchell and Bubenzer, 1984]. Maps are made up of basic agricultural enclaves south of Eastern Siberia [Bazhenova et al., 1997]. And only for the  Tarbagataika we apply other methods of mapping.

Larionov, G.A., The methodology of medium- and small-scale mapping of erosion-hazardous lands, Aktual’nye problemy isucheniya erosii, Moscow, Kolos, 1984, pp. 41-66. [in Russian]

Mitchell, J.K., Bubenzer, G. D., Calculations of soil losses, Soilosion, Moscow, Kolos, 1984, pp. 34-95. [in Russian]

Bazhenova, O.I., Lyubtsova, E.M., Ryzhov, Yu.V., Makarov, S.A., 1997, Spatio-temporal analysis of the dynamics of erosion processes in the south of Eastern Siberia ,. Novosibirsk, Nauka, 208 p. [in Russian]

Lines 126-127 - removed

Line 163 - Climatic fluctuations and their influence on sediment runoff are investigated in detail in other works of the authors (33 and 36). Due to the large volume of the manuscript, climatic variations are not considered here.

Fig. 1 and 2 are changed, corrections are made to the text regarding the links to Fig. 2.

Lines 181-189 - Units - there must be uniformity. On the advice of the reviewer, we will always use tha per year for slopes and tkm2for estimating the flow of suspended sediment. Accumulation rate is usually measured by the height (thickness) of the layer of deluvium or proluvium per year.

Lines 186-187:

20–70 t ha of soil at slopes of 4–6 ° is washed within the Angara basin, depending on the nature of soil cultivation, and 70–200 tha on slopes steeper than 8 ° [39].

Lines 188-190 are better be unchanged:

Most of the washed-out material accumulates at the foot of slopes (from 0.3–2.6 mm/year on deluvial plumes to 55 mm/year below the channel erosion on cropland), in numerous ponds (about 10% of sediment) (but if the reviewer insists, then - such a layer by weight is 3.5 - 30 tha, maximally reaching 630 tha)

Line 101. We used the RUSLE (Revised Universal Soil Loss Equation) model, which takes into account six main factors affecting erosion: climate erosivity, soil erodibility, topographic factors (as slope length and slope steepness), cover management and supporting practices. The efficiency of the model used was largely determined by the availability of baseline information, as the rain erosivity coefficient presented in the Global Rainfall Erosivity Database (GloREDa), compiled from data from 3625 stations from 63 countries, with an average time coverage of 16 was used as climate erosivity , 8 years old (Panagosetal, 2017). Data provided at 30 arc-seconds (~ 1 km) resolution and are available for free download in the European Soil Data Center (ESDAC).

To determine the erodibility of soils (soil erodibility), field data were used on the content of humus, particles of various fractions, soil structure, its stonyness and crushed stone. To calculate topographic factors (LS), we used the ALOS global digital surface model (AW3D30 v. 2.1) with a resolution of 1 ″ (about 30 m), which was obtained by processing optical images from the ALOS satellite operating from 2006 to 2011 (ALOSGlobal ..., 2018). The calculation was made in the open desktop geographic information system SAGA GIS (http://saga-gis.org) according to the formula proposed by Mooreetal (1991):

LS = (n + 1) where

U = upslope contributing area per unit width as a proxy for discharge = Flow Accumulation × Cell Size,

β= slope, n = 0.4, m = 1.3

The topographic factor in the Tarbagataika basin varies from 0 to 40.8 with an average value of 5.93.

As the source cover management data (C-factor) used 30-meter Global Land Cover Dataset (Globeland30) based on 30m multispectral images, including LandsatTM and ETM + and images of Chinese Environmental Disaster Alleviation Satellite. The images acquired over vegetation growing seasons within 2009 - 2010 (National Geomatics ..., 2014). Within Tarbagataika basin land cover is represented by four classes (Table 3).

Since there are no field data on supporting practices in the Tarbagataika basin, the value of the coefficient P was taken as 1, since most of the study area is covered by forest, pastures and deposits of 15-25 years old.

Line 92 - How was slope erosion measured using short-term observations?

Stationary studies make it possible to determine the increase in erosion forms during one shower, during the snowmelt period, during the warm period with showers, over the year, over several years. The method of benchmarks, instrumental surveys, measuring the flow of water and sediment at the top and at the mouth of the gullies was used.

Line 106 how many gullies and what gullies in what landscapes and on what soils were studied Gullies were studied in different regions of the south of Siberia. Gully relief forms are confined to steppe and forest-steppe landscapes. Gullies gravitate towards intermountain hollows, denudation plateaus and river valleys. They cut terraces of rivers and the bottoms of channels of temporary streams. Characterization of quantitative and morphometric characteristics of the gullies in table. 23, p. 82 (Bazhenova et al. Spatially ..., 1997). In the steppes of southern Siberia, the soil cover is represented by ordinary and southern chernozems, as well as chestnut soils, mainly of light mechanical composition.

It seems that in this study there was little actual field monitoring of erosion, which was mainly done using established models and functions related to GIS and other observations.

Line 84 - refers to terrestrial studies, but they are not defined as field studies. I understand that for the assessment in the watershed and on a regional scale, as the authors do, it requires the use of published data and the use of aerial photography and cartographic information to which the models apply. But I would like it to more clearly indicate which specific field measurements or ground tests were carried out to support the modeling approach used. Field studies of erosive landforms by the authors have been carried out since 1974 at the Kharanor physical-geographical field station in the Onon-Argun steppe, since 1995 in the Angara and Baikal regions. We studied the morphometry, morphology and dynamics of erosive landforms using a variety of techniques and instruments, in particular benchmarks, theodalite surveying, etc.

Line 284 — it seems interesting that with a decrease in arable land and livestock grazing, new gullies may arise — is this an assumption that these areas returned to forest communities and that deforestation in them led to an increase in the number of gullies?

In recent years, deforestation has become one of the factors in the formation of gullies. The areas of deforestation have grown significantly. In areas where vegetation and soil cover is disturbed, gullies and gullies occur during heavy rains.

274, 280, 290, etc.: “A large number of erosion forms per unit ... what unit do you mean?

Number of gully forms per 100 km2

Units - removed

307: It’s not clear whether you measured these intensities or if you are referring to a source of literature (which is not mentioned here).

Data on growth rates were obtained by monitoring the growth of more than 150 peaks of gullies in different regions of southern Siberia at physical and geographical stations. (Bazhenova et al. Spatial-temporal analysis of dynamics ..., 1997). Confirmed by the data of other researchers (Ryzhov), (Rysin et al., 2017),

Line 255. It would be helpful to remind the reader of who they are. I suppose they are furrows or drying out waterways?

Gullies

Line 259. I am not familiar with the term valley-balka, but I understand that these are autonomous, disconnected gullies that develop along the river bed.

dic.academic.ru/dic.nsf/bse/115521/Gully

Gully-balka relief is a type of relief characteristic of elevated-plain areas of platform areas, which for a long period of development underwent normal erosion and were dissected by channels of temporary runoff ...

Line 264 - erosion of the slope,  is this gully? Again, using the word scour, which I am not familiar with, is the process of creating a gully.

Rills is better

Line 279 - when you mention 10-25 units, I assume that you determine the number of different types of gullies per square kilometer?

quantity

Line 284 — it seems interesting that with a decrease in arable land and livestock grazing, new gullies may arise — is this an assumption that these areas returned to forest communities and that deforestation in them led to an increase in the number of gullies?

In recent years, deforestation has become one of the factors in the formation of gullies. In areas where vegetation and soil cover is disturbed, rills and gullies occur during heavy rains.

Line 298 - Where are the lakes that are listed? And does the gully network mentioned in the next sentence relate to the deep gullies from the previous sentence?

In Transbaikalia there are listed lakes of Lake Zun-Torei (Geographical coordinates. 49 ° 55’-50 ° 14 ’N, 115 ° 05’-115 ° 98’ E), Lake Bolshoi Chindant (50 ° 6'23 "N 116 ° 24'35" E), Goose Lake (Latitude: 51 ° 06′51 ″ N Longitude: 106 ° 15′41 ″ E). Yes.

Line 301 - V-shaped gullies are young gullies that are still actively developing - are these areas that have recently been taken out of use by humans?

This gullies have a long history of development. In steppe conditions, gullies can retain steep sides for decades.

Line 304 - a sentence starting with "Local and Area ..." is confusing - are these roads that were cut into the landscape in a linear fashion and do they serve as drainage lines?

Local, meaning - single, areal, when several gullies are located next to each other for most of their length.

Line 316 - when you say that the upper horizon of the soil has been blurred, you mean horizon A, and when you mention the crops, I assume that these seeds have not yet sprouted. Then the next sentence assumes that 20 cm tall plants were obtained from the remaining wheat seeds - this is a little confusing.

Plants have already reached a height of 20 cm and were washed during a shower.

Line 318 - what does “fine earth”  mean, is it silt and/or clay in soil with sand, not taken into account?

Particles larger than 1 mm are called the skeletal part, or the skeleton of the soil, less than 1 mm - fine earth.

Line 320 - which means that 37,000 tons of non-productive soils have been redeposited - does this mean that sediment from erosion has deposited on these non-productive soils up the slope?

Non-productive removed

All figures are amended, namely

Comments on Figure 1. We increased the scale bar and the font of coordinates. Signatures of the names of rivers and orographic objects, we consider important, because they carry information about the location of the study area. The background for the picture is a gradient fill created according to the SRTM data and reflecting the terrain. We consider the mention of this in the text of the article unnecessary, because this information, at least in this form, is not used in the study and in this aspect is given only as an illustration.

Comments to Figure 2. Fixed.

Comments on Figure 4. Units have been added to the legend. The black color of the names is chosen because it is the only color that is not in the gradient scale and is the most contrast in this image. For example, white captions on a yellow background will be less contrasting than black ones.

The rest are in text that will be submitted soon

Reviewer 3 Report

Line 49 – is “thou” an acceptable abbreviation for thousand – should it be “thous”

Line 65 – is plowing using a moldboard plow or is discing also involved?

Table 1 – it would be very interesting to know what kind of crops are being grown on the croplands and whether the percentage of each crop changes significantly across and between regions/republics.  Surface soil conditions might be significantly different under different cropping regimes in terms of frequency of soil surface disturbance by cultivation and amount of organic matter left on the surface following harvest.

Line 91 – not clear what is meant by studying surface discharge in the bottoms of temporary streams – does in the bottoms mean the beds of streams and by temporary does that mean ephemeral or intermittent streams?

Line 92 – how was slope erosion measured by short-term observations?

Line 106 – how many gullies and what kind of gullies in what kind of landscape positions and soils were studied

.  It seems that little actual field monitoring of erosion was done in this study – that it was primarily done using established models and functions related to GIS and other observations. In line 84 ground-based investigations are mentioned but not specifically defined as field based. I realize that to make assessments at the watershed and regional scale as the authors are doing requires using published data and use of aerial and mapping scale information to which models are applied.  But I would like to see it more clearly stated as to what specific field measurements or ground truthing was done to back up the modelling approach that was use

Figure 2 It might be good to identify in the figure title that the maximum modulus was assessed for periods of 20-44 years.

Line 169 – I understand the critical importance of the rainfall regime and that may well be the major difference between erosion rates across the different watersheds but little has been said about the kind of soil that exists across the watersheds and soil differences respond differently to the same intensity of rainfall.

Line 173 – I agree that storm water has a higher ability than melted snow to move soil but that is not directly the result of just the water – the same amount of water from a rainfall event as from melting snow on the same slope should move the same amount of soil.  The major difference, I would think, would come from the loosening effect of rainfall impact on the soil surface compared to basically no impact from snow fall.

Line 210 I believe Figure 6 should be Figure 3? 

Line 211 – the sentence suggests high rate of runoff in forested areas of felling and burning but there is no way to identify land cover from Figure 3.  Does one assume that the red areas on the figure are forested or areas of burns and are these burns in the forest or in crop fields?  This is a good figure for identifying erosion runoff but it should be overlayed with a vegetation or land-use map.

Figure 4 – there is no indication of units for the data of this map – I assume that it is tons/ha?  Again a companion figure or overlay of land cover or land use would be very interesting.

Figure 5 – please define drip erosion vs micro-rill erosion.  Is drip erosion the same as splash erosion – the erosion resulting from rain-drop splash?  I am also confused by the use of scour erosion – as I think of erosion rill and gully erosion are both scour erosion as they are the result of water running over the surface that scours soil from small linear depressions that are rills which with further scour result in ephemeral and then classic gullies.  This figure would benefit from contour lines that show the differences in slope of the these three areas.

Line 255 – it would be helpful to remind the reader of who “they” are – I assume they are the rills or ephemeral gullies?

Line 259 – I am not familiar with the term valley=balka but understand that these are stand-alone, non-connected gullies that develop along the channel of a stream.

Line 264 – slope scours are what kind of gully?  Again the use of the word scour that I am familiar with is the process of creating a gully.

Line 279 – when you mention 10-25 units I assume you are identifying the number of different kinds of gullies per square km?

Line 284 – it seems interesting that with the reduction in cropland and grazing pressure that new gullies would arise – is the assumption that these areas returned to forest communities and that the felling in these resulted in the growth rate of gullies?

Figure 7 The gullies in the lower left of A in figure 7 are gullies that developed because of human activity? Is that the what you are saying?  The insert picture needs a B and to me this is not a good picture of a gully – this could be a valley wall along the valley of a meandering stream – is there a chance that the picture could show both sides of the gully instead of just one side?

Line 298 – where are the lakes that are listed?  And the gully net mentioned in the next sentence refers to the highwall gullies in the previous sentence?

Line 301 – the V shaped gullies would suggest young gullies that are still actively down-cutting – are these areas that were recently abandoned by human activity?

Line 304 – the sentence beginning with “Local and areal…” is confusing to me – are these roads that have been cut into the landscape in a linear fashion and do they act as drainage lines?

Line 316 – when you say the upper horizon of the soil was washed away are you referring to the A horizon and when you mention sowings, I assume that if seeds that have not yet germinated.  Is the following sentence then suggesting that the wheat seed that was left only produced plants that were 20 cm in height – this is a bit confusing.

Line 318 – what does “fine earth” refer to is that the silt and/or clay in the soil with the sand left behind?

Line 320 – what is meant by 37 thou t of unproductive soil were resedimented – does that mean these unproductive soils had sediment deposited on them from upslope erosion?

Line 340 – as with the earlier channel you mention analyzing 100 years of channel changes.  What sources of information did you use for the early years of the channel – there probably were no aerial photos of the channel and was there a lot of research focused on stream channel morphology?

Figure 8 – very interesting graphics – can you put dates on the figures or in the title and the shingle-boulder meander bars do not show up very well – the dots are not visible – maybe the legend does not need the dots.

Figure 10 – it is interesting that the lower channel in 10 c is much smaller than it had been in the previous images while the main channel has not seemed to get any larger.  Does that signal that there was a reduction in flow between 1995 and 2017 and if so why?

Line 410 – not sure what a sandy-loamy sandy deposit is – is it a sandy loam or a loamy sand?

Line 427 – What is the reason for all the bank collapse, especially in the forested areas where there are presumably trees down close to the bank edge.  There must be some reason that that reservoir water has risen or there have been more frequent storms that have caused the wave action to cause the banks to collapse.  I would assume the former with rising water and more saturation of the banks there is more potential for bank erosion.  Or is the issue that the reservoirs are relatively new so the whole landscape is become adjust to the new normal for that landscape.

Line 460 – How did a sharp decrease in economic activity influence basin activity – there should be some explanation of what you mean.  Was land not farmed but returned to native plant cover and when you talk about increased river water content do you mean there was more runoff or more chemical content in the water?

I have learned a lot from this paper but it took a lot of effort to learn because many of your international readers will not be familiar with the region of southeastern Siberia.  I would recommend more detailed description of the natural plant communities in each of the watersheds and a more detailed description of the kind of agriculture that is presently and has in the past been practiced in the area.  How large are the crop fields, what kind and what frequency of tillage is used – is plowing mainly mold-board plowing – is it done with tractors or animals.  What specific crops are planted and how are they harvested.  What kind of grazing is done – any rotational grazing is it with cattle, sheep or goats?  I also believe that there is enough information in the paper that the paper could be divided into 2 or 3 papers possibly one for each of the major watersheds.  Also depending on the audience you are writing for a modification of terminology would be helpful.  As I mentioned at the beginning, I believe this is very interesting information that should be published but the paper needs extensive editing and reorganization before it is ready for publicantion.

Author Response

Dear Reviewer,

      The authors are grateful to you for viewing our manuscript and valuable comments aimed at improving the article for publication.

Line 49 –  “thou” is an acceptable abbreviation for thousand

“thou” is an acceptable abbreviation for thousand

Line 65 – is plowing using a moldboard plow or is discing also involved?

According to Table 1, the area of ​​arable land in the south of Eastern Siberia is 5 million hectares. It is not possible to consider all types of tillage, as well as what crops are grown on arable land and whether the percentage of each crop varies significantly in different regions / republics. Its volume does not allow this. In addition, you, as a specialist, are also interested to know that surface soil conditions can vary significantly with different sowing conditions and with different methods of tillage. You are also interested in how much organic matter remains on the surface after harvesting. In our opinion, this is the topic of an independent article. The purpose of our work is to show all possible directions and volumes of substance movement in river basins of the south of Eastern Siberia. According to the authors, such a review will allow not only to show the specific regional features of the fluvial systems of the territory in question, but also to assess the extent of the environmental threat under various scenarios of the development of processes and outline ways to solve problems in order to minimize damage during extreme manifestations of the processes.

A large number of organizations are involved in the aspects of interest to you in the territory under consideration. Among them, it is worth noting the research of the Institute of Soil Science and Agrochemistry of the SB RAS in Novosibirsk, the Institute of Agrarian Problems of Khakassia SB RAS (Abakan), the Altai Research Institute of Agriculture (Barnaul), the Institute of General and Experimental Biology of the SB RAS (Ulan-Ude). Since 1967, after the adoption of well-known laws and regulations aimed at providing urgent measures to protect soils from wind and water erosion, the Altai Research Institute of Agriculture (ANIISH)  began research covering a wide range of erosion, theoretical and applied problems. The work was carried out in close collaboration with leading scientific centers of the country and the region (Problem Laboratory of Soil Erosion of Moscow State University, BelNIIPA, Institute of Soil Science and Agrochemistry, Siberian Branch of the USSR Academy of Sciences, etc.). A wide network of field research and development was deployed experiments involving specialists from about 20 farms in areas of intense manifestation of erosion processes. Active cooperation of ANIISH scientific workers with Kulundinskaya agrostation, educational institutions, specialists of  regional and district departments, farms of the region began to bring their first effects. In the period from 1968 to 1970 subsurface cultivation volumes grew 2.2 times, crosswise slopes processing increased 2.3 times, strip placement of crops and fallow - 8.5 times, sowing with anti-erosion seeders - 2.8 times, tinned highly eroded lands - 1.8 times. Work began on the construction of anti-erosion hydraulic structures. In 1969, 37 km of ramparts were built, in 1970 - 1325 km, peaks of 32 and 117 growing ravines, respectively, were secured. At this time, for the first time in Siberia, methods for contouring tillage on complex slopes were developed and introduced [8]. In the years 1971-1972 the Altai branch of the Rosgiprozem institute, with the participation of ANIISH, developed the “General scheme of a set of soil protection measures on the lands of collective farms and state farms of the Altai Territory”. In the years 1971-1975 volumes of the main agrotechnical anti-erosion measures increased in comparison with 1968-1970 by 1.1-1.7 times, which gave a great economic effect. Crop yields in the whole region increased compared with 1966-1970 by 1.3 times, compared with 1961-1965 by 1.9 times. The Scientific and Technical Council of the Ministry of Agriculture of the RSFSR in October 1975 praised the work done and recommended the experience of the Altai Territory for use in other areas and territories of Siberia. In subsequent years, there has been a further increase in the volume of implementation of agrotechnical anti-erosion measures and the prerequisites have been created for a significant containment of the rate of erosion.

For questions of interest to you, you can see the following publications:

  1. Bazhenova, O.I., Kobylkin, D.V., The dynamics of soil degradation processes within the Selenga basin at the agricultural period, Nat. Resour.,2013, vol. 34, no. 3, pp. 221-227.
  2. Ecology of Russia's erosion-and-river systems,Yu. Belotserkovskii, K.M. Berkovich, O.V. Vinogradova, N.G. Dobrovolskaya, L.V. Zlotina, E.F. Zorina, N.N. Ivanova, Z.P. Kiryukhina, S.N. Kovalev, L.F. Litvin, A.Yu. Sidorchuk, R.S. Chalov, A.V. Chernov; and R.S. Chalov Eds., Moscow, Geograficheskii Fakul’tet MGU, 2002, 163 p.
  3. Korovin, G.N., Nefediev, V.V., Zukert, N.V., and Golovanov, A.S., Retrospective analysis of the forest age-structure, A.S. Isaev Ed., Diversity and dynamics of forest ecosystems in Russia, vol. 1, Moscow, 2012, pp. 16-53.[in Russian]
  4. Lashkin, V.M., Stolyarov, V.I., and Musokhranov, V.E., Soil-protective research in the Altai Territory, Bulletin of the Altai Agrarian State University, 2005, no. 1 (17), pp. 66-72.[in Russian]
  5. Liury, D.I., Goryachkin, S.V., Karavaeva, N.A., Danisenko, E.A., and Nefedova, T.G., The dynamics of agricultural land in Russia in the 20th century and the postagrogenic restoration of vegetation and soils, Moscow, GEOS, 2010, 416 p.
  6. Lobanov, A.I., Savostyanov, V.K., and Pimenov, A.V., Soil Deflation and Agroforestry Measures in the South of Central Siberia (to the 55th anniversary of the organization of the Khakassian antierosion field station at the Sukachev Institute of Forest, SB RAS) , Siberian Forest Journal. 2015. No. 1. P. 105-117.[in Russian]
  7. Namzhilova, L.G., and Tulokhonov, A.K., Evolution of agrarian nature management in Transbaikalia, Novosibirsk, SRC OIGGM, Izd-vo SO RAN, 2000, 200 p.[in Russian]
  8. On the Federal Target Program "Development of Land Reclamation of Agricultural Lands in Russia for 2014-2020",Decree of the Government of the Russian Federation of October 12, 2013, no. 922
  9. Savostyanov, V.K., The development of virgin and fallow lands in Eastern Siberia, Sb. art., dedicated to the 50thanniversary of the development of virgin and fallow lands in Siberia, Novosibirsk, Izd-vo SO RASHN, 2004, pp. 110-118.[in Russian]
  10. Savostyanov, V.K., Use and protection of soils of arid territories of Siberia, RASHN, nauchnoe issledovanie Instituta Agrar. problem Khakassii, the Dokuchaev Soil Science Foundation, Khakas. Branch, Abakan, Kooperativ ‘Zhurnalist’, 2014, 286 p.[in Russian]
  11. Soil resources of the Transbaikalia, Novosibirsk, Izd-vo Nauka, 1989, 182 p.[in Russian]
  12. Tanasienko, A.A., Specific features of soil erosion in Siberia, Novosibirsk, Izd-vo SO RAN, 2003, 175 p.[in Russian]
  13. Tanasienko, A.A., Putilin, A.F., and Artamonova, V.S., Ecological aspects of erosion processes: Analit. Review, (Ser. Ekologiya, issue 55), Izd-vo SPSTL SB RAS, Institute of Soil Science and Agrochemistry of the SB RAS; I. M. Gadzhiyev, Ed., Novosibirsk, 1999, 89 p.[in Russian]
  14. Ubugunov, L.L., Kulikov, A.I., Ubugunova, V.I., Merkusheva, M.G., and Doroshkevich, S.G., Soil Fertility of the Agrolandscapes of Buryatia, Ulan-Ude, Izd-vo BGAA im. Filippova, 2009, 177 p.[in Russian].

Line 91 – not clear what is meant by studying surface discharge in the bottoms of temporary streams – does in the bottoms mean the beds of streams and by temporary does that mean ephemeral or intermittent streams?

Runoff in temporary streams occurs during heavy rains. Water creates new erosion gullies and the development of existing rills and gullies continues.

Line 92 - How was slope erosion measured using short-term observations?

Stationary studies make it possible to determine the increase in erosion forms during one shower, during the snowmelt period, during the warm period with showers, over the year, over several years. The method of benchmarks, instrumental surveys, measuring the flow of water and sediment at the top and at the mouth of the gullies was used.

Line 106 how many gullies and what gullies in what landscapes and on what soils were studied Gullies were studied in different regions of the south of Siberia. Gully relief forms are confined to steppe and forest-steppe landscapes. Gullies gravitate towards intermountain hollows, denudation plateaus and river valleys. They cut terraces of rivers and the bottoms of channels of temporary streams. Characterization of quantitative and morphometric characteristics of the gullies in table. 23, p. 82 (Bazhenova et al. Spatially ..., 1997). In the steppes of southern Siberia, the soil cover is represented by ordinary and southern chernozems, as well as chestnut soils, mainly of light mechanical composition.

It seems that in this study there was little actual field monitoring of erosion, which was mainly done using established models and functions related to GIS and other observations.

Line 84 - refers to terrestrial studies, but they are not defined as field studies. I understand that for the assessment in the watershed and on a regional scale, as the authors do, it requires the use of published data and the use of aerial photography and cartographic information to which the models apply. But I would like it to more clearly indicate which specific field measurements or ground tests were carried out to support the modeling approach used. Field studies of erosive landforms by the authors have been carried out since 1974 at the Kharanor physical-geographical field station in the Onon-Argun steppe, since 1995 in the Angara and Baikal regions. We studied the morphometry, morphology and dynamics of erosive landforms using a variety of techniques and instruments, in particular benchmarks, theodalite surveying, etc.

Рисунок 2. В заголовке рисунка может быть полезно указать, что максимальный модуль был рассчитан для периода от 20 до 44 лет.

Line 169 – I understand the critical importance of the rainfall regime and that may well be the major difference between erosion rates across the different watersheds but little has been said about the kind of soil that exists across the watersheds and soil differences respond differently to the same intensity of rainfall.

Erosion resistance of soils is considered in detail in our collective monograph [Bazhenova et al., 1997].

Line 173 – I agree that storm water has a higher ability than melted snow to move soil but that is not directly the result of just the water – the same amount of water from a rainfall event as from melting snow on the same slope should move the same amount of soil.  The major difference, I would think, would come from the loosening effect of rainfall impact on the soil surface compared to basically no impact from snow fall.

The influence of melt runoff on soil erosion in the south of Eastern Siberia is minimal. Since the water reserves in the snow at the beginning of snow melting are very small, often the snow simply evaporates.

Line 210 I believe Figure 6 should be Figure 3? 

Right, corrected

Line 211 – the sentence suggests high rate of runoff in forested areas of felling and burning but there is no way to identify land cover from Figure 3.  Does one assume that the red areas on the figure are forested or areas of burns and are these burns in the forest or in crop fields?  This is a good figure for identifying erosion runoff but it should be overlayed with a vegetation or land-use map.

Yes, this overlaying was done in Tukhta, S. A., Bazhenova, O. I. and Ryzhov, Yu. V., The Functioning of the Cascade Lithodynamic System of the Kuda River Basin (Upper Angara Region), Geogr. Prir. Resur., 2019, vol. 40, no. 2, pp. 169-179.

Figure 4 – there is no indication of units for the data of this map – I assume that it is tons/ha?  Again a companion figure or overlay of land cover or land use would be very interesting.

Units of measure added to the legend. The figure shows the soil loss in the Tarbagataika basin predicted by RUSLE model, which, inter alia, takes into account soil and land use mosaics.

Рисунок 5 - пожалуйста, уточните, речь идет о капельной эрозии (droperosion) или  мелкоструйчатой эрозии (microrillerosion). Капельная эрозия это тоже самое, что и брызговая эрозия  (splasherosion)- эрозия, вызванная дождевыми брызгами? Меня также смущает использование термина размывание (scourerosion) - так как я думаю, что эрозионные борозды  (erosionrill) и овражная эрозия (gullyerosion) - это и есть размывание (scourerosion), поскольку они являются результатом протекания воды по поверхности, которая размывает почву от небольших линейных впадин, которые представляют собой бороздки, которые с последующим промыванием превращаются в пересыхающие водотоки, а затем в классические овраги. Этот рисунок может быть улучшен контурами, которые показывают различия склонов этих трех областей.

Line 255 – it would be helpful to remind the reader of who “they” are – I assume they are the rills or ephemeral gullies?

gullies

Line 259 – I am not familiar with the term valley=balka but understand that these are stand-alone, non-connected gullies that develop along the channel of a stream.

dic.academic.ru/dic.nsf/bse/115521/Gully

Gully-balka relief is a type of relief characteristic of elevated-plain areas of platform areas, which for a long period of development underwent normal erosion and were dissected by channels of temporary runoff ...

Line 264 – slope scours are what kind of gully?  Again the use of the word scour that I am familiar with is the process of creating a gully.

“Rills” is better instead of “scours”

Line 279 – when you mention 10-25 units I assume you are identifying the number of different kinds of gullies per square km?

quantity

Line 284 – it seems interesting that with the reduction in cropland and grazing pressure that new gullies would arise – is the assumption that these areas returned to forest communities and that the felling in these resulted in the growth rate of gullies?

In recent years, deforestation has become one of the factors in the formation of gullies. In areas where vegetation and soil cover is disturbed, rills and gullies occur during heavy rains.

Fig 7

corrected

Line 298 – where are the lakes that are listed?  And the gully net mentioned in the next sentence refers to the highwall gullies in the previous sentence?

In Transbaikalia there are listed lakes of Lake Zun-Torei (Geographical coordinates. 49 ° 55’-50 ° 14 ’N, 115 ° 05’-115 ° 98’ E), Lake Bolshoi Chindant (50 ° 6'23 "N 116 ° 24'35" E), Goose Lake (Latitude: 51 ° 06′51 ″ N Longitude: 106 ° 15′41 ″ E). Yes.

Line 301 – the V shaped gullies would suggest young gullies that are still actively down-cutting – are these areas that were recently abandoned by human activity?

This gullies have a long history of development. In steppe conditions, gullies can retain steep sides for decades.

Line 304 – the sentence beginning with “Local and areal…” is confusing to me – are these roads that have been cut into the landscape in a linear fashion and do they act as drainage lines?

Local, meaning - single, areal, when several gullies are located next to each other for most of their length.

Line 316 – when you say the upper horizon of the soil was washed away are you referring to the A horizon and when you mention sowings, I assume that if seeds that have not yet germinated.  Is the following sentence then suggesting that the wheat seed that was left only produced plants that were 20 cm in height – this is a bit confusing.

Plants have already reached a height of 20 cm and were washed away during a shower.

Line 318 – what does “fine earth” refer to is that the silt and/or clay in the soil with the sand left behind?

Particles larger than 1 mm are called the skeletal part, or the skeleton of the soil, less than 1 mm - fine earth.

Line 320 – what is meant by 37 thou t of unproductive soil were resedimented – does that mean these unproductive soils had sediment deposited on them from upslope erosion?

Non-productive removed

Line 340 – as with the earlier channel you mention analyzing 100 years of channel changes.  What sources of information did you use for the early years of the channel – there probably were no aerial photos of the channel and was there a lot of research focused on stream channel morphology?

We used the navigation maps of the Lena river published in 1912 and retrospective topographic maps of a scale of 1:84000 published in 1896–1914 (For Irkut-river analyses).

Figure 8 – very interesting graphics – can you put dates on the figures or in the title and the shingle-boulder meander bars do not show up very well – the dots are not visible – maybe the legend does not need the dots.

corrected

Figure 10 – it is interesting that the lower channel in 10 c is much smaller than it had been in the previous images while the main channel has not seemed to get any larger.  Does that signal that there was a reduction in flow between 1995 and 2017 and if so why?

We associate this to the low-water period for the upper reaches of the Angara, the death of the transverse channels and the redistribution of water into the upper river arm (the results of the analysis are being prepared for publication)

Line 410 – not sure what a sandy-loamy sandy deposit is – is it a sandy loam or a loamy sand?

Corrected on “of sandy and loamy deposits”

Line 427 – What is the reason for all the bank collapse, especially in the forested areas where there are presumably trees down close to the bank edge.  There must be some reason that that reservoir water has risen or there have been more frequent storms that have caused the wave action to cause the banks to collapse.  I would assume the former with rising water and more saturation of the banks there is more potential for bank erosion.  Or is the issue that the reservoirs are relatively new so the whole landscape is become adjust to the new normal for that landscape.

Your idea is correct. The level of reservoirs is changing and with rising water levels and greater saturation of the shores, there is greater potential for erosion of the shores. The following study is devoted to this problem: Ovchinikov, G.I., Pavlov, S.Kh. and Trzhitsinskii, Yu.B., Change in Geological Environment in the Influence Zones of the Angara-Yenisei Reservoirs, Novosibirsk, Nauka, 1999, 254 p.

Line 460 – How did a sharp decrease in economic activity influence basin activity – there should be some explanation of what you mean.  Was land not farmed but returned to native plant cover and when you talk about increased river water content do you mean there was more runoff or more chemical content in the water?

The conservation of agricultural land in the Selenga basin has caused the restoration of natural forest vegetation, a decrease in runoff of suspended sediment and runoff of dissolved substances. This led to an improvement in the environmental situation in the river basin and a reduction in pollution of Lake Baikal. The following articles are devoted to this issue:

 Bazhenova, O.I., The Ecological-geomorphological Consequences of Agricultural Lands Within the Lake Baikal Watershed Basin, Geogr. Nat. Resour., 2009, vol. 30, no. 3, pp. 253-257.

Bazhenova, O.I., Kobylkin, D.V., The dynamics of soil degradation processes within the Selenga basin at the agricultural period, Geogr. Nat. Resour., 2013, vol. 34, no. 3, pp. 221-227.

Round 2

Reviewer 2 Report

Dear Olga Bazhenova et al.!

Thank you for your efforts in improving your manuscript and responding to our comments. The presentation of your data has really increased. I do not have any further comments, other than "job well done".

Best regards